# Understanding Calibration Transfer in Knowledge Distillation

## Abstract

Modern deep neural networks are often miscalibrated, leading to overconfident mistakes that erode their reliability, and limit their use in critical applications. The existing confidence calibration techniques range from train-time modification of loss functions to post-hoc smoothing of the classifier's predicted confidence vector. Despite the success of these approaches, it is relatively unclear if supervision from an already trained expert classifier can further enhance a given classifier's confidence calibration. Knowledge distillation (KD) has been shown to help classifiers achieve better accuracy. However, little to no attention has been paid to a systematic understanding if the calibration can also be transferred via KD. In this work, we provide new insights into how and when expert supervision can produce well-calibrated classifiers, by studying a special class of linear teacher and student classifiers. Specifically, we provide theoretical insights into the working mechanisms of KD and show that calibrated teachers can distill calibrated students. We further show that unlike traditional KD where a smaller capacity classifier learns reliably from a larger capacity expert, transfer of calibration can be induced from lower capacity teachers to larger capacity students (aka reverse-KD). Furthermore, our findings indicate that not all training regimes are equally suitable and that a teacher classifier trained using dynamic label smoothing leads to the better calibration of student classifiers via KD. Moreover, the proposed KD based calibration leads to a state-of-the-art (SOTA) calibration framework surpassing all existing calibration techniques. Our claims are backed up by extensive experiments on standard computer vision classification tasks. On CIFAR100 using WRN-40-1 feature extractor, we report an ECE of 0.98 compared to 7.61 and 2.1 by the current SOTA calibration techniques Adafocal (Ghosh et al. (2022)) and CPC Cheng & Vasconcelos (2022) respectively, and 11.16 by the baseline NLL loss (lower ECE is better). The calibration improvement is achieved across various architectures. Using MobileNetv2 on CIFAR100 we report an ECE of 0.88/1.83/4.17/7.76 using Ours/Adafocal/CPC/NLL.

## 1 Introduction

**Calibration.** Deep neural network (DNN) models have become increasingly prevalent in critical applications such as healthcare (Kononenko (2001); Miotto et al. (2018)), and autonomous driving (Bojarski et al. (2016)). In such applications, it is crucial for DNN predictions to not only be accurate but also reliable and trustworthy (Nixon et al. (2019); Dusenberry et al. (2020)). Yet, it has been shown that the softmax probabilities (referred to as *predicted confidence* in this paper) produced by DNNs come with no formal probabilistic guarantees (Guo et al. (2017)). The phenomenon known as *Calibration*, refers to the alignment between a DNN model's predicted confidence and the actual frequency of the event it represents. Calibration indicates model's ability to provide reliable uncertainty estimates, and most modern DNNs are shown to be highly miscalibrated.

**Reasons for Miscalibration and Our Investigation.** Mukhoti et al. (2020) have shown that a DNN model overfitting on NLL loss is the main reason behind highly overconfident predictions, leading to miscalibration. This begs the question, if access to richer class structure and label uncertainties during

training can prevent such overfitting and generate a calibrated classifier. Knowledge distillation (KD) has been used for transferring learnt representations from a (typically large) teacher model to a (usually smaller) student model in multitude of works. In this work we investigate if access to learnt class structure through a teacher model's representation also helps in calibration of a student model.

**Background on Knowledge Distillation.** Since its introduction in 2015 by Hinton et al. (2015), KD has become a go-to method for transferring information between two classifiers with different capacities or architectures. It has been shown repeatedly that student classifiers trained via expert supervision from a teacher classifier or ensemble of classifiers via *soft-label training* exhibit improved performance than when trained with *hard label-based training* (for eg., one hot encoding) via cross-entropy loss Sun et al. (2019); Mirzadeh et al. (2020); Gou et al. (2021). The improved performance is reflected both in terms of increased classification accuracy, as well as stable behavior during training, requiring fewer optimization tricks (Phuong & Lampert (2019)). In summary, student classifiers tend to inherit properties of the teacher classifiers through knowledge sharing between them via KD.

**Our Proposal: KD for Calibration.** Unfortunately, existing explanations of the process of KD rarely go beyond simple qualitative statements attributing improved performance to learning from soft-labels of the expert classifiers. Phuong & Lampert (2019) provide a first theoretical insight into the working mechanism of KD, albeit from an optimization viewpoint. Allen-Zhu & Li (2023) elucidates the effectiveness of ensemble learning and KD in enhancing the test accuracy of classifiers, without placing particular emphasis on the transfer of calibration properties. In our work, we view the role of KD beyond its well-studied role of accuracy transfer and provide theoretical and empirical insights into the transfer of *calibration* to student classifiers. We show, arguably for the first time, that only calibrated teachers potentially distill the best-calibrated students, and thus, a recipe for producing accurate and calibrated classifiers must also involve KD through calibrated teacher classifiers.

**Departure from Current Belief: Does KD Conflict with Calibration.** Interestingly, there is some precedence in investigating the role of calibrating teacher classifiers via label-smoothing (LS) Müller et al. (2019). However, LS was observed to impair KD, i.e., the accuracy of student classifiers degrades when teacher classifiers are calibrated with LS, which potentially points to pitfalls of KD Shen et al. (2021). In our work, we show that this impairment is not the artefact of KD but of the LS itself, which when used to calibrate teacher classifiers and distill their understanding to student classifiers at higher temperatures, ends up over-smoothing a student's predictions, thereby significantly degrading its accuracy (Chandrasegaran et al. (2022)). We show that teachers trained via dynamic label-smoothing methods (E.g., Hebbalaguppe et al. (2022b)) consistently distil calibrated students across all temperatures. To this end, we highlight the role of KD in calibrating classifiers and argue strongly in favor of using knowledge sharing from calibrated experts to student classifiers as the most promising calibration technique.

**Mathematical Definition.** Formally, given a data distribution $\mathcal{D}$ of $(x, y) \in X \times \{0, 1\}$ and let $c$ be the predictive confidence, the predictor $f : X \to [0, 1]$ is said to be calibrated (Dawid (1982)), if:

$$\mathbb{E}_{(x,y)\sim\mathcal{D}} \big[y \mid f(x) = c\big] \; = \; c, \quad \forall c \in (0, 1) \tag{1}$$

**Key Contributions.** To achieve calibration of DNNs, we bring together two seemingly unrelated subfields: KD and confidence calibration. We make the following key contributions in this direction:

1. ***Understanding calibration transfer via distillation***: We develop a theoretical framework to analyze KD and its ability to transfer learning of a teacher classifier to a student classifier, and show, arguably for the first time, that *only* calibrated teachers can distill calibrated students. We corroborate our theoretical results later through exhaustive experiments.

2. ***Achieving best student network calibration***: Our experiments demonstrate that students trained via KD from teachers that are first calibrated using dynamic/adaptive label-smoothing, exhibit the best calibration compared to other train-time/post-hoc calibration techniques. (Sec. 5.1). Our framework is dubbed **KD(C)** (**K**nowledge **D**istillation from **C**alibrated teacher).

3. ***Not all calibration techniques are compatible with KD***: It has been observed empirically that LS impairs KD (Müller et al. (2019)). This impairment is argued to be a high-temperature phenomenon (Chandrasegaran et al. (2022)). In our experiments, we, too, observe a similar behavior when

teacher classifiers are trained via static calibration methods, such as `LS`. However, we show that when the teacher classifiers are calibrated using dynamic `LS` methods the distillation produces calibrated student classifiers consistently across wide temperature regimes.

4. ***Calibration distillation works both ways***: Contrary to popular belief that only larger models can effectively distill their learning to smaller models, we show that smaller calibrated models can also yield better calibrated larger models. The observation is consistent with our key insight that the availability of label ambiguities through soft-labels during training is extremely useful for calibration. This setting is relevant when large calibrated models or large datasets for training such large models not readily available and significantly widens the applicability of our framework.

## 2 RELATED WORK

Research pertaining to calibration of `DNNs` can be categorized into following approaches: (a) train-time calibration, (b) post-hoc calibration, (c) Bayesian inference, and (d) data augmentation. A ***train-time calibration*** integrates model calibration during the training phase through suitable modification of loss function. For instance, label smoothing (`LS`), originally introduced by Szegedy et al. (2015) to improve the classifier accuracy by computing cross-entropy with a weighted sum of one-hot vector and the uniform distribution, was adopted by Müller et al. (2019) for improving calibration. Most train-time methods for calibration improvement inherently look to smooth confidence scores in a sample-agnostic manner. Noteworthy works in this category include those by Moon et al. (2020); Kim et al. (2021); Liu et al. (2022); Hebbalaguppe et al. (2022b); Park et al. (2023).

Conversely, ***post-hoc calibration*** focuses on optimizing calibration measures using a separate hold-out set post-training. Guo et al. (2017), demonstrated that temperature scaling (`TS`), a technique that smooths confidence scores by dividing the logits of a classifier by a scalar $T > 1$, enhances its calibration. Other notable contributions in this category also include the studies by Platt et al. (1999); Kull et al. (2017; 2019); Bohdal et al. (2021); Islam et al. (2021). However, despite its simplicity, it was observed in Hebbalaguppe et al. (2022b) that train-time approaches offer superior performance over post-hoc methods. Prominent examples of calibration methods relying on data augmentation encompass Thulasidasan et al. (2019) and Hebbalaguppe et al. (2022a). Meanwhile, Bayesian methodologies are exemplified by Gal & Ghahramani (2016); Lakshminarayanan et al. (2017); Ovadia et al. (2019) and Wenzel et al. (2020). However, in the context of our research, train-time and `KD`-based approaches are especially pertinent.

**KD** Hinton et al. (2015) was originally proposed to enhance the accuracy of student classifiers by transferring knowledge from high-capacity teacher classifiers. However, recent empirical evidence points to regularization effects of `KD` over student classifiers akin to training classifiers separately via `LS` (see Tang et al. (2020)), which seems to suggest direct calibration benefits of `KD`. It was shown in Yuan et al. (2021) that when the temperature parameter during `KD` is set to unity and the probability distribution of teacher classifiers are assumed to be uniform, `KD` via teacher classifier and `LS` of student classifier exhibit identical behaviors in terms of gradient propagation. The observations prompted to explore the scenario where a teacher classifier itself is first calibrated via `LS` and then distills knowledge to a student classifier (i.e., distill knowledge from an `LS` calibrated teacher) with the hope of doubling the regularization benefits. However, it was observed in Müller et al. (2019) that `LS` representations impaired with those of `KD`, thus nullifying any regularization benefits. This viewpoint, nonetheless, was later shown to be incomplete by Shen et al. (2021), where the authors argued that such an impairment is only a high-temperature phenomenon. While this interplay between `LS` and `KD` provided some insights into the regularization benefits of `KD`, its role as a potential calibrator of smaller teacher classifiers has not been addressed in the literature. In our work, we look beyond just the vanilla `LS` of teacher classifiers and provide direct theoretical and empirical evidence towards the benefits of working with calibrated teachers and how they distill `SOTA` calibrated student classifiers via `KD`. We also systematically analyze various calibration techniques for the teacher classifiers so that the resulting student classifiers exhibit significantly improved calibration performance over directly calibrating them via train-time or post-hoc methods. We show that *dynamic* `LS` methods, such as the `MDCA` (Hebbalaguppe et al. (2022b)), consistently exhibit better accuracy and calibration trade-off across wider temperature ranges.

## 3 UNDERSTANDING CALIBRATION VIA KNOWLEDGE DISTILLATION

We analyze the mechanics of obtaining calibrated models via KD from a theoretical standpoint, focusing on linear teacher and student networks in a binary classification problem. Such linear classifiers, which were initially explored in Phuong & Lampert (2019) to gain a general understanding of KD, have not been previously investigated for their potential to transfer learned representations, particularly calibration, to student networks. Furthermore, the authors in Phuong & Lampert (2019) utilized a simplified version of the KD loss function, which did not consider the significance of distillation weights and quadratic temperature scaling. These factors play a crucial role in showcasing the transfer of calibration to student models.

To this end, we represent an $i^{\text{th}}$ training instance by $\mathbf{x}_i \in \mathbb{R}^d$. The set of all training examples is represented by $\mathbf{X} \in \mathbb{R}^{d \times N}$. We use $z_{i,s}$ and $z_{i,t}$ to represent logits of the student and teacher networks for the $i^{\text{th}}$ training instance, respectively. These logits can be converted into valid probability distributions $p_{i,s}$ and $p_{i,t}$, respectively, using Sigmoid activation function. In knowledge distillation, the output probabilities of the teacher network are softened using inverse temperature scaling of the logit, leading to prediction probabilities $\{\tilde{p}_{i,t}\}$. The true class labels are denoted by $\{y_i \in \{0, 1\}\}$. Since, the teacher and student networks are assumed to be linear networks, an arbitrary deep network can equivalently be represented using a single layer network. We use $\mathbf{W}_s$ and $\mathbf{W}_t$ to represent weight matrices of the student and teacher networks, respectively. Finally, we use $T \in \mathbb{R}_+$ to depict temperature parameter for temperature scaling, while $\alpha \in [0, 1)$ represents the relative importance of the student's binary cross-entropy loss. Below we list the key assumptions before presenting the key theoretical results.

**Assumption 1.** The feature dimension $d$ is larger than the number of training examples $N$.

**Assumption 2.** The student and teacher networks are represented by linear networks.

**Remark 1.** A direct consequence of Assumption 1 is that the data matrix $\mathbf{X}$ is full column rank almost surely, since if one randomly samples $N$ training examples (with $N < d$), the probability that any two sample vectors are perfectly aligned is nearly zero. Consequently, the matrix $\mathbf{X}^\top \mathbf{X}$ is invertible. Assumption 2 ensures that both student and teacher networks can be compactly represented as single layer linear networks. Though, the assumption implicitly enforces student network to be of the same capacity as that of the teacher network, it makes it easier to better understand the mechanics of the KD, specifically when we later distill larger models from smaller models. Extending to nonlinear networks poses significant challenges, similar to the lack of a general theory for DNNs and non-convex optimization. However, we can still extract valuable insights from the theory of linear networks and utilize them to establish a framework applicable to general nonlinear networks.

In KD, the student aims to minimize the weighted combination of the binary cross-entropy loss $\mathcal{L}_{\text{BCE}}$, and the KD loss $\mathcal{L}_{\text{KD}}$, given by:

$$
\mathcal{L}_{\text{BCE}} = -\sum_{i=1}^{N} \left[ y_i \log p_{i,s} + (1 - y_i) \log (1 - p_{i,s}) \right],
$$
$$
\mathcal{L}_{\text{KD}} = -T^2 \sum_{i=1}^{N} \left[ \tilde{p}_{i,t} \log \tilde{p}_{i,s} + (1 - \tilde{p}_{i,t}) \log (1 - \tilde{p}_{i,s}) \right],
$$
$$
\mathcal{L}_{\text{tot}} = (1 - \alpha)\mathcal{L}_{\text{BCE}} + \alpha \mathcal{L}_{\text{KD}}, \tag{2}
$$

where $p_{i,s} := \sigma(\mathbf{W}_s^\top \mathbf{x}_i)$, $\tilde{p}_{i,s} := \sigma\left(\dfrac{\mathbf{W}_s^\top \mathbf{x}_i}{T}\right)$, $\tilde{p}_{i,t} := \sigma\left(\dfrac{\mathbf{W}_t^\top \mathbf{x}_i}{T}\right)$ and $\sigma(\cdot)$ is the Sigmoid function. Below we provide our key theoretical results.

**Theorem 1.** Let $\mathbf{X} \in \mathbb{R}^{d \times N}$ be a data matrix satisfying Assumption 1, and $\mathbf{W}_s$ and $\mathbf{W}_t$ represent the parameters of the student and the teacher networks, respectively. Then, under Assumption 2 and using the gradient-descent algorithm, the parameters $\mathbf{W}_s$ of the student network converge to:

$$
\mathbf{W}_s \approx \alpha \mathbf{W}_t + 4(1 - \alpha)\mathbf{X}(\mathbf{X}^\top \mathbf{X})^{-1}\mathbf{Y}_{1/2},
$$

where $\mathbf{Y}_{1/2} := \left[ y_i - \frac{1}{2} \right]_{i=1}^{N}$ is an $N$-dimensional vector.

*Proof.* Please refer to Section 2.1 in the supplementary material for the detailed proof. ∎

**Remark 2.** Theorem 1 shows that when $\alpha$ is close to unity, the weights of the student network are almost identical to those of the teacher network. Thus, properties of the teacher network transfer directly to the student. For $\alpha \neq 1$, the student also learns to update its weight from the labeled data.

**Calibrated Teachers produce Calibrated Students.** A neural network classifier is said to be well calibrated if the predicted probability distribution is similar to the observed probability distribution (Naeini et al. (2015)). Mathematically speaking, if a teacher network with predicted probabilities $\{p_{i,t}\}$ is well calibrated, then the following holds:

$$\sum_{i=1}^{N} p_{i,t} = \sum_{i=1}^{N} y_i. \tag{3}$$

We now prove that well-calibrated teachers distill well-calibrated students. On the contrary, if the teacher classifier is not well-calibrated, it is impossible to distill well-calibrated student classifiers. The result extends our understanding of KD beyond accuracy transfer and formally characterizes the transfer of calibration from student-to-teacher networks.

**Theorem 2.** Let Assumptions 1-2 hold. Let $t_c$ and $t_{uc}$ be two teacher classifiers with output probabilities $\{p_{i,t_c}\}$ and $\{p_{i,t_{uc}}\}$, respectively. Also, let $s_c$, $s_{uc}$ depict two student classifiers trained independently from the corresponding teacher classifiers $t_c$ and $t_{uc}$ through KD, with output probabilities $\{p_{i,s_c}\}$ and $\{p_{i,s_{uc}}\}$, respectively. Furthermore, assume that the teacher classifier $t_c$ is well calibrated, then the student classifier $s_c$ is also well calibrated. Conversely, if the teacher classifier $t_{uc}$ is uncalibrated, the corresponding student classifier $s_{uc}$ mimics a similar behavior, i.e.,

$$\sum_{i=1}^{N} p_{i,s_c} = \sum_{i=1}^{N} y_i, \quad \text{and} \quad \sum_{i=1}^{N} p_{i,s_{uc}} \neq \sum_{i=1}^{N} y_i.$$

*Proof.* Please refer to Section 2.2 in the supplementary material for detailed proof. ∎

Below we describe our framework for calibrating classifiers that leverage KD as a key tool.

## 3.1 Recipe for Joint Optimization of Calibration and Accuracy

Our main goal is to create well-trained DNNs that demonstrate the highest possible accuracy and confidence calibration during inference. Traditional loss functions like negative log-likelihood (NLL) have been found to encourage over-confident models. To overcome the shortcomings of NLL-based training, one potential but simplistic approach involves calibrating classifiers pre-trained via KD, with the hope that these models will retain enhanced accuracy from the teacher model (refer to Theorem 1) and achieve confidence calibration through subsequent post-hoc calibration. However, our experiments illustrate that this two-step approach does not guarantee the best results. We hypothesize

| Abbreviation | Description of Calibration Technique |
|---|---|
| NLL | Negative Log Liklihood |
| LS Szegedy et al. (2015) | Label Smoothing |
| CE+TS Guo et al. (2017) | Temperature scaling on student model trained with cross-entropy loss |
| MMCE Kumar et al. (2018) | Maximum Mean Calibration Error |
| MixUp Thulasidasan et al. (2019) | Calibration through MixUp data augmentation |
| CRL Moon et al. (2020) | Correctness Ranking Loss |
| PSKD Kim et al. (2021) | Progressive refinement in Self Knowledge Distillation |
| MDCA Hebbalaguppe et al. (2022b) | Multi-class difference in Confidence and Accuracy |
| AdaFocal Ghosh et al. (2022) | Adaptive Focal Loss |
| CPC Cheng & Vasconcelos (2022) | Calibration via Pairwise Constraints |
| MbLS Liu et al. (2022) | Margin-based Label Smoothing |
| KD(UC) | Knowledge Distillation from Uncalibrated teacher |
| **KD with TS (Ours)** | Temperature Scaling on distilled Student classifier |
| **KD with LS (Ours)** | KD framework with teacher trained on LS |
| **KD with MDCA (Ours)** | KD framework with teacher trained on MDCA |
| **KD with AdaFocal (Ours)** | KD framework with teacher trained on AdaFocal |
| **KD with CPC (Ours)** | KD framework with teacher trained on CPC |
| **KD with CRL (Ours)** | KD framework with teacher trained on Correctness Ranking Loss |
| **KD with MbLS (Ours)** | KD framework with teacher trained on Margin based Label Smoothing |
| **KD with MixUp (Ours)** | KD framework with teacher trained on MixUp data augmentation |

Table 1: Nomenclature used for baseline methods and the corresponding description of calibration technique. Our proposed approach KD(C) variants are highlighted where 'C' is calibration type

that the suboptimal performance of the student classifier primarily arises from the interference between the representations learned through KD and the post-hoc calibration technique.

As shown in Theorem 2, calibrated teachers are guaranteed to distill calibrated students. Hence, we experiment with KD using calibrated teachers which can potentially distill accurate and calibrated students in one go. For calibrating teacher networks, we restrict our attention to train-time methods, such as MDCA (Hebbalaguppe et al. (2022b)) and LS (Szegedy et al. (2015)), since post-hoc methods

| Architecture / Method | WideResNet-40-1 (0.56M) | | | | MobileNetV2 (2.25M) | | | |
|---|---|---|---|---|---|---|---|---|
| | Top1 (%) ↑ | ECE (%) ↓ | SCE (%) ↓ | ACE (%) ↓ | Top1 (%) ↑ | ECE (%) ↓ | SCE (%) ↓ | ACE (%) ↓ |
| NLL | 70.04 | 11.16 | 0.30 | 11.19 | 66.09 | 7.76 | 0.25 | 7.80 |
| LS Szegedy et al. (2015) | 70.07 | 1.30 | 0.21 | 1.49 | 66.96 | 4.24 | 0.23 | 4.18 |
| CE+TS Guo et al. (2017) | 70.04 | 2.57 | **0.19** | 2.50 | 66.09 | 2.33 | **0.19** | 2.37 |
| MMCE Kumar et al. (2018) | 69.69 | 7.34 | 0.25 | 7.37 | 62.90 | 3.21 | 0.21 | 3.15 |
| MixUp Thulasidasan et al. (2019) | 72.04 | 2.57 | 0.21 | 2.52 | 67.53 | 8.69 | 0.28 | 9.73 |
| CRL Moon et al. (2020) | 65.80 | 13.91 | 0.37 | 13.91 | 67.05 | 12.06 | 0.33 | 12.06 |
| PSKD Kim et al. (2021) | **72.56** | 3.73 | 0.20 | 3.72 | 69.09 | 6.95 | 0.23 | 6.94 |
| MDCA Hebbaluguppe et al. (2022b) | 68.51 | 1.35 | 0.21 | 1.34 | 66.96 | 1.61 | 0.20 | 1.92 |
| AdaFocal Ghosh et al. (2022) | 67.36 | 2.10 | 0.21 | 1.97 | 65.34 | 1.83 | 0.20 | 1.53 |
| CPC Cheng & Vasconcelos (2022) | 69.99 | 7.61 | 0.23 | 7.55 | 67.30 | 4.17 | 0.22 | 4.07 |
| MbLS Liu et al. (2022) | 69.97 | 5.37 | 0.22 | 5.37 | 67.22 | 1.25 | 0.20 | 1.25 |
| KD(UC) | 69.60 | 15.18 | 0.37 | 15.18 | 66.82 | 5.40 | 0.22 | 5.36 |
| **Ours (KD with MixUp)** | 72.48 | 1.21 | 0.20 | 1.17 | **69.92** | 2.17 | 0.24 | 2.10 |
| **Ours (KD with AdaFocal)** | 71.70 | 1.19 | **0.19** | 1.34 | 66.64 | 1.55 | 0.20 | 1.43 |
| **Ours (KD with CPC)** | 70.00 | 9.02 | 0.26 | 9.01 | 67.83 | **0.88** | **0.19** | **0.95** |
| **Ours (KD with MDCA)** | 71.07 | **0.98** | 0.20 | **1.10** | 67.17 | 1.10 | 0.20 | 1.17 |

Table 2: **[Large-to-small]** Comparison of calibration performance of small student models calibrated using KD(C) framework vs. SOTA calibration techniques, employing a relatively larger calibrated teacher on CIFAR100 dataset. WRN-40-2 (Zagoruyko & Komodakis (2016)) and ResNeXt-18x4 (Xie et al. (2017)) were used as teachers for WRN-40-1 (Zagoruyko & Komodakis (2016)) and MobileNetV2 (Sandler et al. (2018)) respectively as student models. For ECE/SCE computation, 15 bins were used in accordance with prior work. ACE uses an adaptive binning strategy. For full results refer to the supplementary . **Numbers in bold:** best performance; underlined: second best. KD(C) gives best all around performance, in one instance PSKD is slightly better than KD(C) in accuracy, but lags much behind in calibration metric (the focus of this paper).

are known to result in relatively inferior calibration performance (Müller et al. (2019); Platt et al. (1999); Kull et al. (2019)). Supported by the theoretical insights of generic property transfer from a teacher to a student network, we describe our training regime in two steps:

1. ***Train-time calibration of teachers***: We draw inspiration from train-time calibration techniques that have shown superior performance than post-hoc calibration, and have experimented with the following techniques: (Kumar et al. (2018); Müller et al. (2019); Hebbalaguppe et al. (2022b); Cheng & Vasconcelos (2022); Moon et al. (2020)) to name a few. A simple gradient analysis reveals that train-time calibration methods, such as MDCA (Hebbalaguppe et al. (2022b)) and ACLS (Park et al. (2023)), act as *dynamic/adaptive* label smoothing, which is arguably better than the traditional *static* label smoothing (Müller et al. (2019)).

2. ***Knowledge distillation from calibrated teacher***: Once trained for calibration and accuracy, teacher classifiers distill their behavior to student classifiers through KD loss Eq. (2). As a result, the student classifiers are both accurate and confidence calibrated (see Theorem 2).

Proposed comprehensive framework KD(C) encompasses the full spectrum, enabling models with varying capacity (smaller/larger) to distill student models with the least calibration error and better accuracy compared to the SOTA post-hoc/train-time calibration methods. **Note:** We do not advocate a specific train-time calibrator but rather a KD-style calibration where an expert model helps enhance the calibration performance of a student. Tab. 1 shows the KD(C) framework encompasses the variations highlighted in Cyan. With this, we study systematically the effect of various direct calibration methods. The best results among "KD with" methods are shown in each table. Please refer to the supplementary for a full version of these results.

## 4 EXPERIMENTS

We now validate our theoretical claims on calibrated teachers distilling calibrated students (see Theorem 2 in supplementary material) through extensive experiments.

**Evaluation Metrics:** We benchmark our framework KD(C) against other competing methods using **(a)** calibration error metrics (lower value is better), Expected calibration error (ECE) (Guo et al.

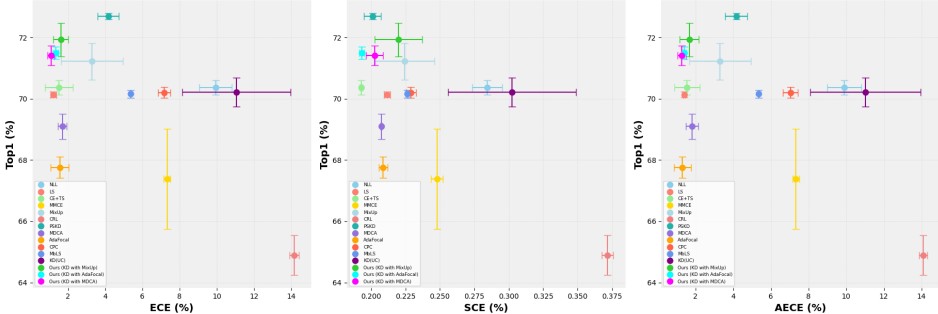

Figure 1: **Comparative study of accuracy vs. calibration trade-offs associated with existing calibration techniques and ours (Top-left is most preferred)**: The mean and one standard scatter error bars for `Top1`, `ECE` and `SCE` of `WideResNet-40-1` trained on `CIFAR100` using `SOTA` calibration techniques. `WideResNet-40-2` was used as Teacher for KD(UC) and the proposed, KD(C) variants. Note: KD(C) variants (magenta, cyan, and green) achieve the best results in terms of `ECE`, `ACE` and `SCE`, along with slight boosts in `Top1` (an inherent KD-property). Further, the lower variances emphasize the reliability of KD(C) variants. All plots were generated by training `WideResNet-40-1` models through every calibration technique on 3 runs.

(2017)), Static Calibration Error (`SCE`) and Adaptive Calibration Error (`ACE`) (Nixon et al. (2019)), as well as **(b)** Top1 accuracy (higher value is better), indicative of generalization performance.

**Datasets and Baselines.** We use widely accepted diverse datasets, `CIFAR10` (Krizhevsky et al. (2009)), `CIFAR100` (Krizhevsky et al. (2009)), and Tiny-ImageNet (Le & Yang (2015)) for benchmarking. To test robustness of our approach, we report additional results on `CIFAR100-C` (Hendrycks & Dietterich (2019)) dataset in the supplementary. We could not experiment on ImageNet due to limited computational constraints in our lab. We include models trained through standard `NLL`, as well as `LS`, `MixUp`, `Adafocal`, `MMCE`, `CRL`, `CPC`, `MDCA`, and `PSKD` (please refer to Tab. 1 for citations on calibration techniques). Along with this, we include student KD(UC) distilled from an uncalibrated teacher obtained by training using `NLL` as one of the baselines.

**Training details.** The architectures used in the experiments include `ResNet`(He et al. (2016)), `MobileNetV2` (Sandler et al. (2018)), and `ShuffleNetV2` (Ma et al. (2018)) `DenseNet` (Huang et al. (2018)), `WideResNet`(Zagoruyko & Komodakis (2016)) architectures. The exact details of training and model hyperparameters, along with the details on compute resources are included in the supplementary material. Source code /trained models for all benchmark methods will be made public upon acceptance for reproducibility.

## 5 RESULTS

**Large calibrated teacher models distilling into smaller models.** We now present compelling evidence supporting the superiority of our proposed KD(C) method over the `SOTA` train-time and post-hoc techniques for calibrating smaller student classifiers. To this end, we leverage distillation to create a smaller model (e.g., `WRN-40-1`/`MobileNetV2`) from a well-calibrated teacher model (e.g., `WRN-40-2`) and compare its performance with models directly subjected to train-time calibration techniques, as well as the progressive-KD (`PSKD`) method introduced by Kim et al. (2021). We also report the impact of distillation from an uncalibrated teacher model, denoted as KD(UC), as a baseline. The summarized results are detailed in Tab. 2. Notably, KD(C) demonstrates significantly lower calibration errors (`ECE`/`SCE`/`ACE`) while simultaneously achieving higher accuracy compared to models calibrated directly using metrics such as `NLL`/`MDCA`/`LS`. Additionally, Fig. 1 provides a visual representation of our findings, illustrating the mean and standard deviations of accuracy and calibration errors over three random runs. Notably, KD(C) variants exhibit (a) the best balance between accuracy and calibration while (b) displaying higher reliability, as evidenced by their lower variance. Importantly, our results confirm our theoretical findings discussed in Sec. 3, establishing

that calibrated teachers are capable of effectively distilling calibrated students. This underscores the successful transfer of learned representations, encompassing both accuracy and calibration aspects, from a calibrated teacher model to a smaller student. We give additional results in supplementary showing our approach consistently yields improved calibration across various model architectures. Reliability diagrams corresponding to Tab. 2 can also be found in the supplementary.

**Self-distillation.** A significant question that arises pertains to the generalizability of insights gleaned from the prior set of experiments. Particularly whether these insights can be extended to produce accurately calibrated classifiers with identical architecture and capacity. Our research demonstrates that this process referred to as "self-distillation," results in classifiers that exhibit superior calibration compared to their teachers. However, the increase in accuracy is only marginal, likely due to the absence of distillation from a teacher with greater capacity. Our findings on the CIFAR-10 dataset are succinctly presented in Tab. 3. It is worth noting that, unlike the PSKD approach proposed by Kim et al. (2021), which progressively distills knowledge by leveraging

| Calibration Method | Top1 (%) ↑ | ECE (%) ↓ | SCE (%) ↓ | AECE (%) ↓ |
|---|---|---|---|---|
| NLL | 89.87 | 3.30 | 0.75 | 3.28 |
| LS Szegedy et al. (2015) | 89.60 | 7.10 | 1.78 | 6.75 |
| CE+TS Guo et al. (2017) | 89.90 | 0.96 | **0.40** | 0.77 |
| MMCE Kumar et al. (2018) | 89.38 | 1.20 | 0.51 | 0.94 |
| MixUp Thulasidasan et al. (2019) | 89.57 | 9.42 | 2.07 | 9.41 |
| CRL Moon et al. (2020) | **90.31** | 2.92 | 0.72 | 2.81 |
| PSKD Kim et al. (2021) | 89.21 | 3.27 | 0.93 | 3.25 |
| MDCA Hebbalaguppe et al. (2022b) | 88.74 | 0.99 | 0.46 | 0.80 |
| AdaFocal Ghosh et al. (2022) | 88.98 | 0.79 | 0.44 | 0.86 |
| CPC Cheng & Vasconcelos (2022) | 89.26 | 3.47 | 0.79 | 3.44 |
| MbLS Liu et al. (2022) | 89.86 | 2.83 | 0.69 | 2.78 |
| KD(UC) | 89.88 | 0.99 | 0.43 | 0.82 |
| **Ours (KD with TS)** | 90.23 | 0.51 | 0.41 | 0.59 |
| **Ours (KD with MixUp)** | 89.27 | 9.16 | 2.19 | 9.05 |
| **Ours (KD with AdaFocal)** | 89.56 | 0.63 | 0.41 | 0.65 |
| **Ours (KD with CPC)** | 89.92 | 0.64 | 0.48 | 0.67 |
| **Ours (KD with MDCA)** | 88.79 | **0.48** | 0.48 | **0.54** |
| **Ours (KD with MMCE)** | 89.97 | 0.85 | 0.54 | 0.84 |

Table 3: **[Self-distillation]** using MobileNetV2 feature extractor on CIFAR10 dataset. Only top-3 performing KD(C) variants are reported, for full results refer to supplementary.

aging the previous epoch's trained model, KD(C) employs self-distillation just once with a fixed teacher throughout the training process, following a methodology akin to Zhang & Sabuncu (2020). Nevertheless, KD(C) achieves remarkable improvements in terms of calibration errors, notably surpassing the performance of the baseline models by a significant margin.

**Small calibrated teacher models distilling into large models.** In settings where large trained models are not available, it is desirable to be able to distill the knowledge from smaller models to larger models. Jiang & Deng (2023) have shown that smaller models can also be valid teachers for large students, however, it was observed that the gains in accuracy were not significant as compared to distilling from a large teacher comparatively. Our results for the configuration are summarized in Tab. 4, where smaller model MobileNetV2 is used as teacher network to calibrate DenseNet-121. We note a trade-off between accuracy and calibration performance, primarily arising from the

| Method | Top1 Acc.(↑) | ECE(↓) | SCE(↓) | ACE(↓) |
|---|---|---|---|---|
| NLL | 94.81 | 3.37 | 0.72 | 3.34 |
| LS Szegedy et al. (2015) | 94.33 | 4.58 | 0.85 | 4.56 |
| CE+TS Guo et al. (2017) | 94.81 | 0.97 | 0.40 | 1.23 |
| MMCE Kumar et al. (2019) | 93.08 | 1.01 | 0.39 | 0.92 |
| MixUp Thulasidasan et al. (2019) | **95.18** | 2.85 | 0.67 | 2.83 |
| CRL Moon et al. (2020) | 93.67 | 1.36 | 0.45 | 1.25 |
| PSKD Kim et al. (2021) | 94.49 | 1.81 | 0.43 | 2.02 |
| MDCA Hebbalaguppe et al. (2022b) | 92.69 | **0.31** | 0.35 | **0.35** |
| AdaFocal Ghosh et al. (2022) | 93.48 | 1.44 | 0.43 | 1.10 |
| CPC Cheng & Vasconcelos (2022) | 94.39 | 4.36 | 0.91 | 4.18 |
| MbLS Liu et al. (2022) | 94.45 | 3.42 | 0.75 | 3.43 |
| KD(UC) | 90.2 | 2.17 | 0.60 | 2.13 |
| **Ours (KD with AdaFocal)** | 91.68 | 0.54 | **0.34** | 0.61 |
| **Ours (KD with MDCA)** | 90.09 | 0.53 | 0.45 | 0.51 |
| **Ours (KD with MbLS)** | 93.1 | 0.61 | 0.38 | 0.40 |

Table 4: **[small-to-large]** Calibration performance of a large student model (DenseNet-121)(6.95M) on CIFAR10 when distilled from small (un)calibrated teacher (MobileNetV2(2.25M)). Top3 performing KD(C) variants have been shown. See to supplementary for additional results.

inherent capacity differences between larger student models and smaller-capacity teachers. Larger student models possess greater capacity and can potentially match or surpass the performance of smaller-capacity teachers. Consequently, this limits the additional knowledge that can be effectively distilled from smaller-capacity teachers. Nevertheless, the process still yields improvements in calibration for larger models through KD.

**Iterative self-distillation.** Taking inspiration from previous works such as Mirzadeh et al. (2020); Yalburgi et al. (2020); Kim et al. (2021), we investigate whether KD(C) can iteratively distill more accurate and calibrated models. Here both teacher and student have identical architectures, and a

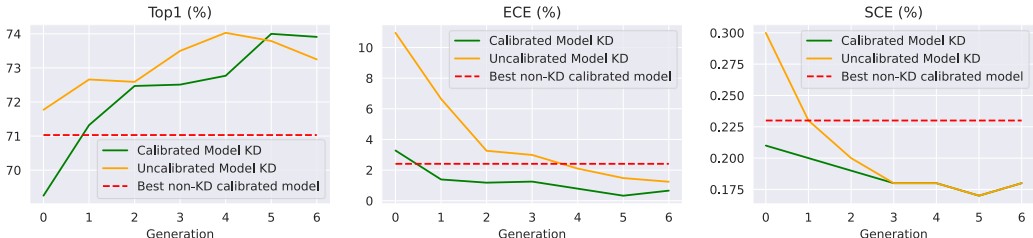

Figure 2: Iterative self-KD on `CIFAR100` using ResNet56. We use `KD` with `MDCA` for calibration.

student in t[th] iteration (called *generation* hereon) becomes teacher for (t+1)[th] generation. We refer to this as iterative self-distillation. Fig. 2 shows the self-distillation process for six generations. As expected the gap between KD(UC) and KD(C) gradually diminishes with each generation (the only difference between the two is initialization: generation zero teacher is uncalibrated in KD(UC) but calibrated in KC(C)). Our observation aligns with findings from (Zhang & Sabuncu (2020); Kim et al. (2021)).

**Note:** From Tables 2, 3, and 4, we observe no other method is as consistent in improving calibration performance as KD(C); notice that while direct calibration can be the best-performing in one of the instances, it fails miserably in other instances. The KD(C) framework variants are consistently the best-performing ones or the second-best.

**Other Experiments.** We report **(a)** calibration performance under dataset drift on vision datasets; **(b)** ablation study on the effect of hyper-parameters like $T$ (temperature) and $\alpha$ (distillation weight) in the supplementary along with experiments involving other `DNN` architectures.

## 5.1 DISCUSSION

Our work offers promising results with a recipe to combine `KD` and calibration in one go. Unlike traditional `LS`, dynamic/adaptive smoothing based regularizers (Park et al. (2023); Hebbalaguppe et al. (2022b)) offer sample-specific dynamic label-smoothing. However, not many of these methods tend to capture inter-class semantics, which are inherently captured through the interplay of teacher-based knowledge transfer and learning directly from data. Inspired from Chandrasegaran et al. (2022), in the supplementary, we give additional rationale for the superior performance of the proposed KD(C) framework using penultimate layer visualizations. We also provide more justification on why teacher classifiers need to be calibrated at train-time in the supplemental material.

## 6 CONCLUSIONS

Our primary contribution lies in providing a robust theoretical foundation, including formal proofs, for the transfer of calibration and accuracy between teacher and student `DNN` models. We have demonstrated consistent superior calibration achieved through our KD(C) method. Our experiments span diverse scenarios, encompassing large-to-small, small-to-large, and self-distillation settings, featuring a variety of architectures and datasets. The consistent success of KD(C) across these different configurations highlights its effectiveness and broad applicability. Additionally, we advocate for the adoption of dynamic calibration techniques (such as `MDCA`) on the teacher model before the distillation. Our findings are further enriched by the insights gained from penultimate layer visualizations, shedding light on the inner workings of calibration in `DNN`s. From an application perspective, as the utilization of `KD` continues to grow, particularly for obtaining lightweight models beneficial for edge computing, neural architecture search, model compression, and other domains, our work contributes by introducing a method to additionally enhance the trustworthiness of these neural networks through calibration.

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

# Supplementary Material: Understanding Calibration Transfer in Knowledge Distillation

**Anonymous authors**

## Contents

# 1 INTRODUCTION

We complement our main text with supplementary materials encompassing the following components:

1. **Theoretical Insights**: This section contains main theoretical results, along with an explanation of the rationale behind choosing Padé approximants over more commonly used Taylor approximation, as discussed in Lemma 1.

2. **Rationale for Enhanced Performance**: This section elucidates the superior performance of the KD(C) framework, attributing it to three key factors: (a) insights from penultimate visualizations, (b) considerations of inter-class semantic similarities, and (c) the careful design of calibrators for the teacher model.

3. **Illustration of Generality**: Included is Fig. S5, which provides a visual demonstration of KD(C)'s versatility by comparing direct calibration with the KD(C) framework. It also presents an example featuring the Hebbalaguppe et al. (2022) regularizer.

4. **Expanded Experimental Scope**: We strengthen the KD(C) methodology with additional experiments, covering various scenarios, including large-to-small, small-to-large, self-distillation, and iterative self-distillation. These experiments involve different descriptors and datasets.

5. **Additional Results**: We provide supplementary results that encompass calibration performance in the presence of dataset drift and reliability diagrams featuring confidence histograms, as elaborated in Sec. 5.1.

6. **Hyperparameter Analysis**: A detailed study explores how calibration and accuracy are influenced by various hyperparameters in the KD(C) framework, as depicted in Fig. S8.

7. **Source Code**: The supplementary materials include the source code along with a `readme.md` file, enclosed within the provided zip file.

8. **Training and Compute Details**: We furnish comprehensive information on the specifics of training and compute resources employed in our experiments.

9. **Limitations and Broader Impact**: This section delves into the limitations of our research and contemplates its broader impact on the field.

These supplementary materials serve to enrich and provide a deeper understanding of our main findings and contributions.

# 2 THEORETICAL SUPPORT: ADDITIONAL DETAILS

## 2.1 PROOF OF THEOREM 1

The proof of Theorem 1 is contingent on several essential Lemmas, which will be introduced beforehand. Lemma 1 and Lemma 2 capture the effect of quadratic temperature scaling in the KD loss function, $\mathcal{L}_{\text{KD}}$. In particular, it is shown that the partial derivative of $\mathcal{L}_{\text{KD}}$ w.r.t. student's logit for a given sample is equal to the difference in predicted probabilities of the student and teacher classifiers for that sample. These results are leveraged to characterize the first-order condition of optimality for the total loss function $\mathcal{L}_{\text{tot}}$ w.r.t. parameters of the student classifier.

**Lemma 1.** *Let $z_{i,s} := \mathbf{W}_s^\top \mathbf{x}_i$ and $z_{i,t} := \mathbf{W}_t^\top \mathbf{x}_i$ with $\tilde{p}_{i,s}$, $\tilde{p}_{i,t}$ be defined as above. Then $\lim_{T \to \infty} T(\tilde{p}_{i,s} - \tilde{p}_{i,t}) \approx p_{i,s} - p_{i,t}$.*

*Proof.* The result follows as a consequence of Padé approximation. Recall that from definition,

$$\tilde{p}_{i,s} - \tilde{p}_{i,t} = \frac{1}{1 + e^{-z_{i,s}/T}} - \frac{1}{1 + e^{-z_{i,t}/T}} \approx \frac{1}{1 + \frac{1 - \frac{z_{i,s}}{2T}}{1 + \frac{z_{i,s}}{2T}}} - \frac{1}{1 + \frac{1 - \frac{z_{i,t}}{2T}}{1 + \frac{z_{i,t}}{2T}}}, \qquad \text{(S1)}$$

where the last approximation follows from Padé approximation of the exponential when $T$ is large.

Thus, Eq. (S1) can be re-written as:

$$\tilde{p}_{i,s} - \tilde{p}_{i,t} = \frac{1 + z_{i,s}/2T}{2} - \frac{1 + z_{i,t}/2T}{2} \implies T(\tilde{y}_{i,s} - \tilde{y}_{i,t}) = \frac{z_{i,s} - z_{i,t}}{4}. \qquad \text{(S2)}$$

On the other hand, a similar analysis following the Padé approximation yields:

$$p_{i,s} - p_{i,t} \approx (z_{i,s} - z_{i,t})/4. \qquad \text{(S3)}$$

Thus, Lemma 1 follows directly from Eq. (S2) and Eq. (S3). $\qquad\square$

**Remark:** Padé approximants have a wider range of convergence than the corresponding Taylor series, and can even converge where the Taylor series does not. For a detailed exposition, please refer to Sec. 2.3 and Fig. S1.

The following result shows that the quadratic temperature scaling in the KD loss function ensures that the gradients used to update the network weights are independent of the smoothed labels.

**Lemma 2** (Quadratic temperature scaling). *Let $\mathcal{L}_{KD}$ be defined as in Eq. (2). Then,*

$$\lim_{T \to \infty} \frac{\partial \mathcal{L}_{KD}}{\partial z_{i,s}} = p_{i,s} - p_{i,t}.$$

*Proof.* Recall that by definition $\tilde{p}_{i,s} = \dfrac{1}{1 + e^{-z_{i,s}/T}}$. The partial derivative of $\tilde{p}_{i,s}$ w.r.t. $z_{i,s}$ reads:

$$\frac{\partial \tilde{p}_{i,s}}{\partial z_{i,s}} = \frac{1}{T}\tilde{p}_{i,s}(1 - \tilde{p}_{i,s}). \qquad \text{(S4)}$$

On the other hand,

$$\frac{\partial \mathcal{L}_{\text{KD}}}{\partial z_{i,s}} = -T^2 \left( \frac{\tilde{p}_{i,t}}{\tilde{p}_{i,s}} - \frac{1 - \tilde{p}_{i,t}}{1 - \tilde{p}_{i,s}} \right) \frac{\partial \tilde{p}_{i,s}}{\partial z_{i,s}} = T^2 \frac{(\tilde{p}_{i,s} - \tilde{p}_{i,t})}{\tilde{p}_{i,s}(1 - \tilde{p}_{i,s})} \frac{\partial \tilde{p}_{i,s}}{\partial z_{i,s}}. \qquad \text{(S5)}$$

Thus, from Eq. (S4), Lemma 1 and for large $T$, Eq. (S5) reduces to:

$$\lim_{T \to \infty} \frac{\partial \mathcal{L}_{\text{KD}}}{\partial z_{i,s}} = p_{i,s} - p_{i,t},$$

which completes the proof. $\qquad\square$

**Lemma 3.** *The derivative of the total loss function $\mathcal{L}_{tot}$ w.r.t. the parameters $\mathbf{W}_s$ of the student network lies in the span of $\mathbf{X}$, and is given by:*

$$\frac{\partial \mathcal{L}_{tot}}{\partial \mathbf{W}_s} = \sum_{i=1}^{N} (p_{i,s} - \{\alpha p_{i,t} + (1 - \alpha)y_i\}) \, \mathbf{x}_i.$$

*Proof.* The proof follows directly from Lemma 2. $\qquad\square$

**Theorem 1.** *Let $\mathbf{X} \in \mathbb{R}^{d \times N}$ be a data matrix satisfying Assumption 1, and $\mathbf{W}_s$ and $\mathbf{W}_t$ represent the parameters of the student and the teacher networks, respectively. Then, under Assumption 2 and using the gradient-descent algorithm, the parameters $\mathbf{W}_s$ of the student network converge to:*

$$\mathbf{W}_s \approx \alpha \mathbf{W}_t + 4(1 - \alpha)\mathbf{X}(\mathbf{X}^\top \mathbf{X})^{-1}\mathbf{Y}_{1/2},$$

*where $\mathbf{Y}_{1/2} := \left[ y_i - \frac{1}{2} \right]_{i=1}^{N}$ is an $N$-dimensional vector.*

*Proof.* First observe that the minimum value of the total loss function in Eq. (2) is finite. Moreover, the total loss function is convex in the parameters of the student network. Thus, any gradient-based descent algorithm with suitable step-size will converge to the optimizer asymptotically fast.

We now characterize the set of optimizers. Recall that the first-order condition of optimality implies:

$$\frac{\partial \mathcal{L}_{\text{tot}}}{\partial \mathbf{W}_s} = 0 \implies \sum_{i=1}^{N} (p_{i,s} - \{\alpha p_{i,t} + (1 - \alpha)y_i\}) \, \mathbf{x}_i = 0,$$

where the last equality follows from Lemma 3. Since the vectors $\{\mathbf{x}_i\}$ are linearly independent (see Remark 1), the above equality holds if:

$$p_{i,s} - \{\alpha p_{i,t} + (1-\alpha)y_i\} = 0, \ \ \forall i \in \{1, \ldots, N\}. \tag{S6}$$

Expanding Eq. (S6) in terms of logits $z_{i,s}$ leads to:

$$\frac{1}{1+e^{-z_{i,s}}} = \alpha \frac{1}{1+e^{-z_{i,t}}} + (1-\alpha)y_i \implies \frac{1}{1+\frac{1-\frac{z_{i,s}}{2}}{1+\frac{z_{i,s}}{2}}} \approx \alpha \frac{1}{1+\frac{1-\frac{z_{i,t}}{2}}{1+\frac{z_{i,t}}{2}}} + (1-\alpha)y_i, \tag{S7}$$

where the last equation follows from Padé approximation. Rearranging the terms in Eq. (S7), and using the fact that $z_{i,s} = \mathbf{W}_s^\top \mathbf{x}_i$ and $z_{i,t} = \mathbf{W}_t^\top \mathbf{x}_i$, one obtains:

$$\left(\mathbf{W}_s - \alpha \mathbf{W}_t\right)^\top \mathbf{x}_i = 4(1-\alpha)\left(y_i - 1/2\right).$$

Since the above condition holds for every $i \in \{1, \ldots, N\}$, the vector form of it can be written as:

$$\mathbf{X}^\top(\mathbf{W}_s - \alpha \mathbf{W}_t) = 4(1-\alpha)\mathbf{Y}_{1/2}, \tag{S8}$$

which is an underdetermined system of linear equations whose least-norm solution is given by:

$$\mathbf{W}_s = \alpha \mathbf{W}_t + 4(1-\alpha)\mathbf{X}(\mathbf{X}^\top\mathbf{X})^{-1}\mathbf{Y}_{1/2}, \tag{S9}$$

which completes the proof. $\square$

## 2.2 PROOF OF THEOREM 2

**Theorem 2.** *Let Assumptions 1-2 hold. Let $t_c$ and $t_{uc}$ be two teacher classifiers with output probabilities $\{p_{i,t_c}\}$ and $\{p_{i,t_{uc}}\}$, respectively. Also, let $s_c$, $s_{uc}$ depict two student classifiers trained independently from the corresponding teacher classifiers $t_c$ and $t_{uc}$ through KD, with output probabilities $\{p_{i,s_c}\}$ and $\{p_{i,s_{uc}}\}$, respectively. Furthermore, assume that the teacher classifier $t_c$ is well calibrated, then the student classifier $s_c$ is also well calibrated. Conversely, if the teacher classifier $t_{uc}$ is uncalibrated, the corresponding student classifier $s_{uc}$ mimics a similar behavior, i.e.,*

$$\sum\nolimits_{i=1}^{N} p_{i,s_c} = \sum\nolimits_{i=1}^{N} y_i, \ \ and \ \ \sum\nolimits_{i=1}^{N} p_{i,s_{uc}} \neq \sum\nolimits_{i=1}^{N} y_i.$$

*Proof.* From Eq. (S6), the first-order condition for optimality for a student $s$ trained from a teacher $t$ through KD reads:

$$\sum\nolimits_{i=1}^{N} p_{i,s} = \alpha \sum\nolimits_{i=1}^{N} p_{i,t} + (1-\alpha) \sum\nolimits_{i=1}^{N} y_i,$$

which can be rewritten as

$$\sum_{i=1}^{N}(p_{i,s} - y_i) = \alpha \sum_{i=1}^{N}(p_{i,t} - y_i).$$

Thus for the same value of $\alpha \in (0,1)$, if the teacher classifier $s_c$ is well calibrated, then

$$\sum_{i=1}^{N}(p_{i,s_c} - y_i) = \alpha \left(\sum_{i=1}^{N}(p_{i,t_c} - y_i)\right) = 0,$$

where, the last equality follows from well calibration of the teacher classifier. On the other hand,

$$\sum\nolimits_{i=1}^{N}(p_{i,s_{uc}} - y_i) = \alpha \sum\nolimits_{i=1}^{N}(p_{i,t_{uc}} - y_i) \neq 0 \implies \sum\nolimits_{i=1}^{N} p_{i,s_{uc}} \neq \sum\nolimits_{i=1}^{N} y_i,$$

which completes the proof. $\square$

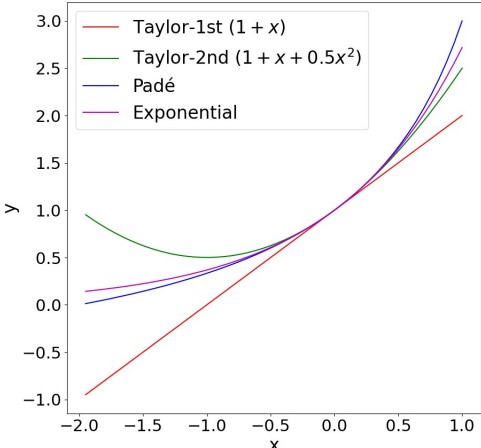

Figure S1: Padé vs Taylor for a simple exponential function. Note that Padé approximants offer superior reliability compared to the extensively used Taylor approximants.

## 2.3 PADÉ VS TAYLOR APPROXIMANTS

Employing approximants to derive theoretical outcomes in `DNNs` is commonplace due to the intricacies of dealing with highly nonlinear equations. We illustrate the difference between Padé and Taylor's approximation as follows: Padé approximants have a wider range of convergence than the corresponding Taylor series, and can even converge where the Taylor series does not. A simple example of Padé approximant is, $e^x = \frac{e^{0.5x}}{e^{-0.5x}} \approx \frac{(1+0.5x)}{(1-0.5x)} = (1+0.5x)(1-0.5x)^{-1}$, which for $|x| < 2$ can further be expanded to $e^x \approx (1+0.5x)(1-0.5x)^{-1} = (1+0.5x)(1+0.5x+0.25x^2+\dots)$. Thus, despite using first-order approximations for both the numerator and denominator terms, the above Padé approximant very closely follows the original exponential function. This is in contrast to Taylor's expansion, and even a second-order Taylor's expansion does not mimic the exponential function, except for a very small interval around the origin. Please refer to Figure S1 for further details.

This is precisely why we restrict using Padé approximants in our theoretical exploration since they are still potentially non-divergent in regimes even when $z_{i,t}$ and $z_{i,s}$ are not vanishingly small. It must also be remarked that **exact characterization of weights of student network is a theoretically hard problem**, and such practical approximations are useful to obtain important theoretical insights.

## 3 RATIONALE ON THE SUPERIOR PERFORMANCE OF OUR KD(C) FRAMEWORK

The essence of Theorem 2 lies in its assertion that uncalibrated teachers can only transfer their lack of calibration to their student counterparts, whereas calibrated teachers enable the distillation of calibrated students. This theorem underscores the crucial significance of utilizing calibrated teachers in the knowledge distillation process. In light of this observation, we advocate for a novel approach to achieving accurate and calibrated models: calibrating a model through distillation from another model that is already calibrated. To validate the efficacy of this approach, we conducted an extensive series of experiments, showcasing the capabilities of our framework, KD(C). Our experimental results provide compelling evidence that KD(C) yields student models characterized by two key attributes: dynamic calibration at the sample level and semantic calibration. These findings substantiate the effectiveness of our proposed framework in achieving both sample-level and semantic calibration in student models.

### 3.1 CLASSIFICATION OF LABEL SMOOTHING

**Standard/static label smoothing.** Label Smoothing (`LS`) serves as a regularization technique designed to address potential inaccuracies within datasets. It recognizes that maximizing the likelihood directly, denoted as $P(y|\mathbf{x})$, may be detrimental due to the possibility of errors in the training labels.

To mitigate this issue, `LS` introduces controlled noise into the labeling process. In essence, `LS` operates as follows: Given a small constant value $\epsilon$, it considers the training label $y$ to be correct with a probability of $(1 - \epsilon)$ and incorrect otherwise. Specifically, in the context of a softmax model with $k$ outputs, it replaces the traditional binary classification targets of $0$ and $1$ with modified targets. These modified targets consist of $\frac{\epsilon}{(k-1)}$ for incorrect labels and $(1 - \epsilon)$ for correct labels Szegedy et al. (2015); Müller et al. (2019). This approach ensures that all output probabilities undergo uniform regularization, thereby helping to combat overfitting and improve model generalization.

**Adaptive Label Smoothing.** In this method the level of regularization applied to training labels, which are typically one-hot encoded, is dynamically adjusted based on the network's output probabilities for different classes Cheng & Vasconcelos (2022); Hebbaluguppe et al. (2022); Park et al. (2023). This method is found to be more beneficial than conventional static label smoothing (`LS`) proposed in Szegedy et al. (2015).

**Conditional Label Smoothing.** In this method the training labels go through selective modifications based on specific criteria, such as the application of margin-based penalties Liu et al. (2022). This approach places its emphasis on and applies regularization solely to the probabilities that exhibit miscalibration, thereby demonstrating enhanced calibration capabilities.

## 3.2 Visualization of penultimate layer's activations reveal KD(C) using dynamic regularization works better than static regularization

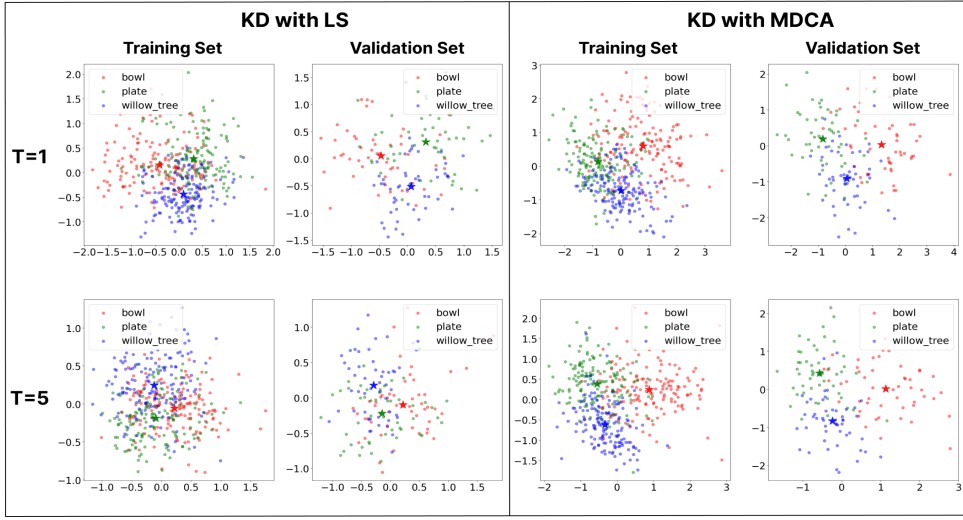

Figure S2: **Visualization of penultimate layer's activations (Teacher = ResNet56, Student = ResNet8, Dataset = CIFAR100)**. We train `ResNet8` using calibration techniques: `KD with LS` (Left column) and `KD with MDCA` (Right Column). We follow the same setup and procedure used in papers [Müller et al. (2019); Shen et al. (2021)] We use two semantically similar classes (`bowl`, `plate`) and one semantically dissimilar class (`willow_tree`). A '*' in the plot for each cluster represents its cluster's centroid. A well-calibrated teacher can effectively capture the inter-class relationships and serve as a reliable dynamic label smoothing prior such as `MDCA` Hebbaluguppe et al. (2022). Observe that the classes: `bowl` and `plate` are visually similar and hence the penultimate visualizations of these classes should be closer than the dissimilar class: `willow_tree`. As the temperature $T$ is increased the similar classes diffuse into one in the case of `KD with LS` while `KD with MDCA` offers better separation, retaining the semantic similarity while being well separated from the dissimilar class.

**Penultimate Visualization.** Müller et al. (2019) introduced this visualization technique wherein they projected the penultimate activations onto the hyperplane defined by the template vectors (weight vectors) corresponding to the selected classes (three classes) for visualization.

**Systematic diffusion.** The concept of "systematic diffusion," introduced by Chandrasegaran et al. (2022), was developed to address discrepancies observed in prior studies, particularly the contra-

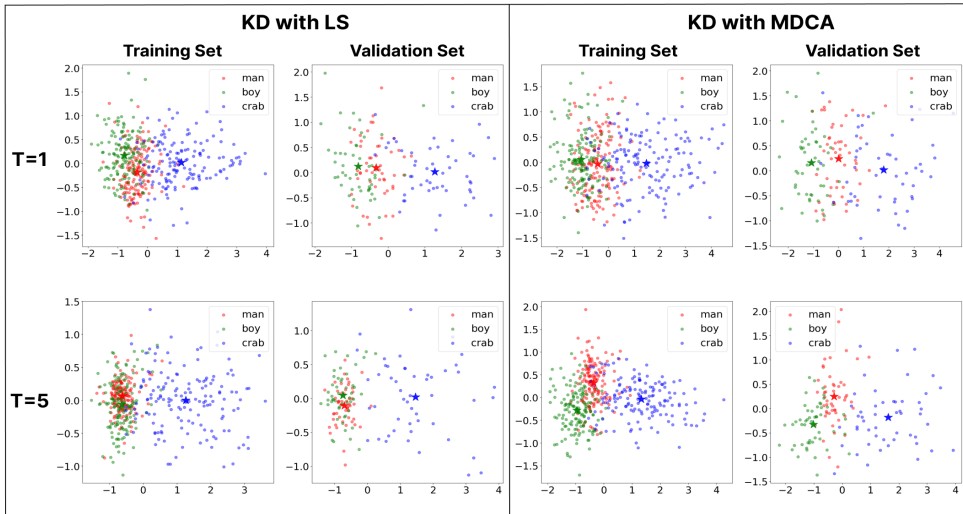

Figure S3: **Visualization of penultimate layer's activations (`Teacher = ResNet56`, `Student = ResNet8`, `Dataset = CIFAR100`)**. We train `ResNet8` using calibration techniques: `KD with LS` (Left column) and `KD with MDCA`(Right Column). We follow the same setup and procedure used in papers [Müller et al. (2019); Shen et al. (2021)] We use two semantically similar classes (`man`, `boy`) and one semantically dissimilar class (`crab`). A '*' for each cluster represents its cluster's centroid. A well-calibrated teacher can effectively capture the inter-class relationships and serve as a reliable dynamic label smoothing prior such as `MDCA` Hebbalaguppe et al. (2022). Observe that the classes: `man` and `boy` are visually similar and hence the penultimate visualizations of these classes should be closer than the dissimilar class: `crab`. As the temperature $T$ is increased the similar classes diffuse into one in the case of `KD with LS` while `KD with MDCA` offers better separation, retaining the semantic similarity while being well separated from the dissimilar class.

dictions between Shen et al. (2021) and the insights presented in `LS` literature Müller et al. (2019). This concept aims to elucidate the compatibility of label smoothing with knowledge distillation. The findings from Chandrasegaran et al. (2022)'s work indicate that when `KD` is conducted at elevated temperatures from a teacher model trained with `LS`, it results in a systematic shift in the relationships between classes. Specifically, for semantically similar classes, the inter-cluster distance decreases, while for the remaining classes, it increases relatively. Importantly, this diffusion of classes is not random; rather, it follows a systematic pattern.

In Fig. S2 and Fig. S3, we provide visual evidence of the limitations associated with `LS`-trained teachers compared to `MDCA` teachers Hebbalaguppe et al. (2022). These Penultimate layer visualizations, inspired by the work of Shen et al. (2021), reveal that semantically similar classes experience systematic diffusion when using `LS`, whereas this phenomenon is not observed with `MDCA` calibration. This observation substantiates our recommendation to opt for dynamic smoothing regularization techniques such as `MDCA`.

Notably, we notice a trend where distilled student models are most calibrated when the distillation temperature ($T$) is approximately 1. We hypothesize that increasing $T$ leads to the destruction of discriminating features, as outlined by Chandrasegaran et al. (2022), due to systematic diffusion among highly similar classes as seen in the penultimate representations. These discriminating features are crucial for achieving calibration by resolving confusion among similar classes. However, as $T$ increases further, we simultaneously amplify the relationships between somewhat related classes Tang et al. (2020), while diminishing the relationships between very similar classes. This nuanced understanding highlights the intricate interplay between temperature, class relationships, and calibration, shedding light on the optimal conditions for achieving calibration in `KD` scenarios.

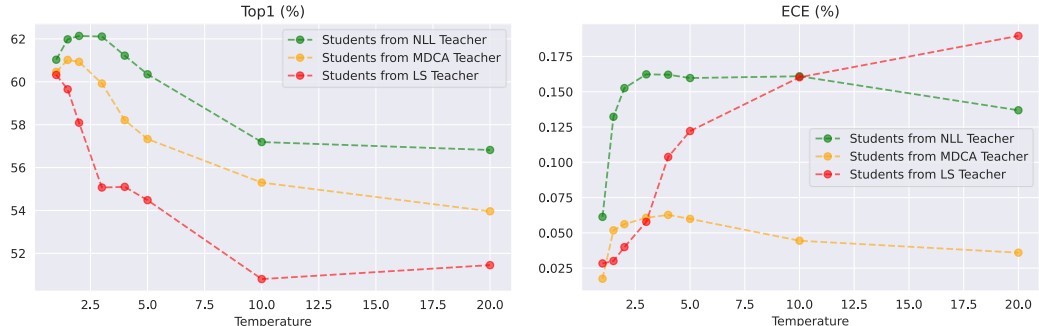

Figure S4: **[Study of `ECE` variablity in case of KD(C), specifically we consider `KD with LS` and `KD with MDCA` and study variation of accuracy and calibration as a function of temperature]**: Comparison of Top $1\%$ accuracy and `ECE` when train-time calibration method is changed from Label Smoothing Szegedy et al. (2015) and `MDCA` Hebbalaguppe et al. (2022): We use `ResNet56` teacher on `CIFAR100` and distill to `ResNet8`. Note that `MDCA`-based students have lower accuracy than `NLL`, however, `ECE` is largely stable when temperature $T$ is varied.

## 4 ILLUSTRATION OF THE GENERALITY OF KD(C) FRAMEWORK

Fig. S5 presents a novel framework KD(C) that leverages calibrated teachers through `KD` to produce `DNNs` with the least calibration error. This comprehensive framework encompasses the full spectrum, enabling models with varying capacity (smaller/larger) to distill student models with the least calibration error and better accuracy compared to the SOTA post-hoc/train-time calibration methods.

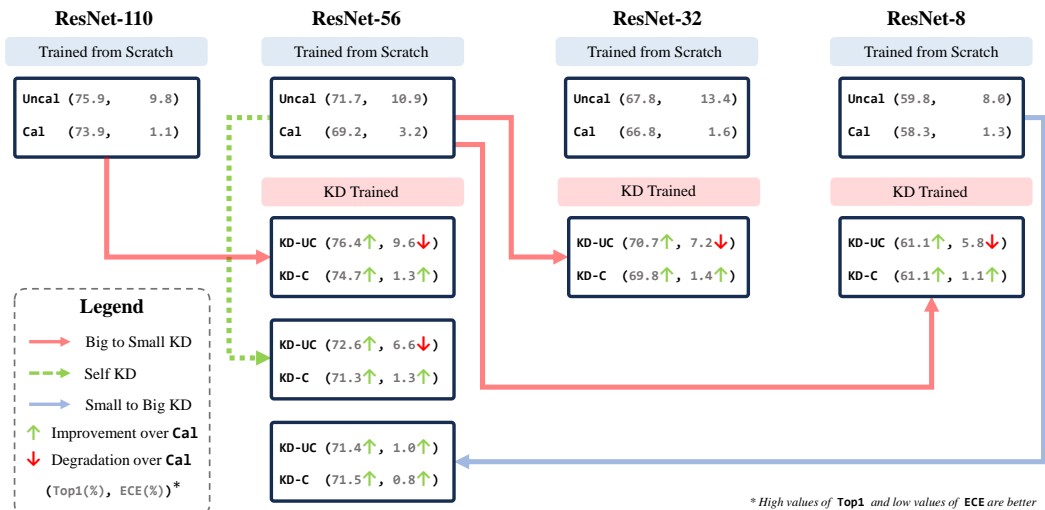

Figure S5: **An illustration of KD(C) framework's generality using calibration method as MDCA**. We can distill a calibrated student from a large teacher and vice-versa yielding SOTA calibration without any trade-offs in accuracy. "Uncal" and "Cal" mean uncalibrated and calibrated teachers trained using `NLL` and a recent `SOTA` calibration technique Hebbalaguppe et al. (2022) respectively. KD(UC) and KD(C) refer to students distilled using "Uncal" and "Cal" teachers respectively. Going from a large calibrated teacher to a smaller student yields `SOTA` calibrated student, with an additional boost in accuracy (E.g., compare `ResNet56` "Trained from scratch" with `ResNet56` "KD-trained" student from ResNet110). Self-distillation and going from a smaller teacher to a bigger student also have a similar effect on calibration, however, the gains in accuracy are comparable to respective models trained from scratch. The above results are on `CIFAR100`.

## 5 ADDITIONAL RESULTS

### 5.1 RELIABILITY DIAGRAMS AND CONFIDENCE HISTOGRAMS

Reliability diagrams serve as effective visual aids for assessing the calibration of `DNNs`. They involve partitioning the predicted probabilities generated by `DNNs` into a predetermined number of bins along the $x$-axis. The $y$-axis represents the normalized count of events (e.g., class = "dog") within each bin. A well-calibrated model will exhibit points that closely align with the main diagonal, spanning from the bottom left to the top right of the plot. Reliability diagrams corresponding to Fig. S6 are included to show that KD(C) variants obtain near `SOTA` results.

### 5.2 EFFECT OF HYPER-PARAMETERS LIKE $T$ (TEMPERATURE) AND $\alpha$

We investigate the influence of hyperparameters $T$ and $\alpha$ on both calibrated and uncalibrated teacher models, as visually depicted in Fig. S7 (big-to-small) and Fig. S8 (small-to-big).

**(a) Big teacher, Small student**: In this scenario as we increase the value of $\alpha$, we witness an intuitive rise in calibration. However, this effect is predominantly noticeable for small values of $T$ (depicted in the bottom-right region of Fig. S7). Generally, the calibration errors (`ECE`) incurred by distilling students from a calibrated teacher tend to be markedly lower than those distilled from an uncalibrated teacher, as evident from the bottom row in Fig. S7. **(b) Small teacher, Big student**: Initially, we observe an expected trend: as $\alpha$ increases (signifying a higher dependence on the teacher), accuracy experiences a decrease. This outcome arises from the process of distillation from a weaker teacher. However, when distilling from a calibrated teacher, we discern that elevating $\alpha$ results in enhanced calibration. Nevertheless, this improvement in calibration is accompanied by a trade-off with accuracy.

Notably, we find that optimal calibration is generally achieved when $T \approx 1$, regardless of the size of the teacher model employed. This observation aligns with the findings presented in Stanton et al.

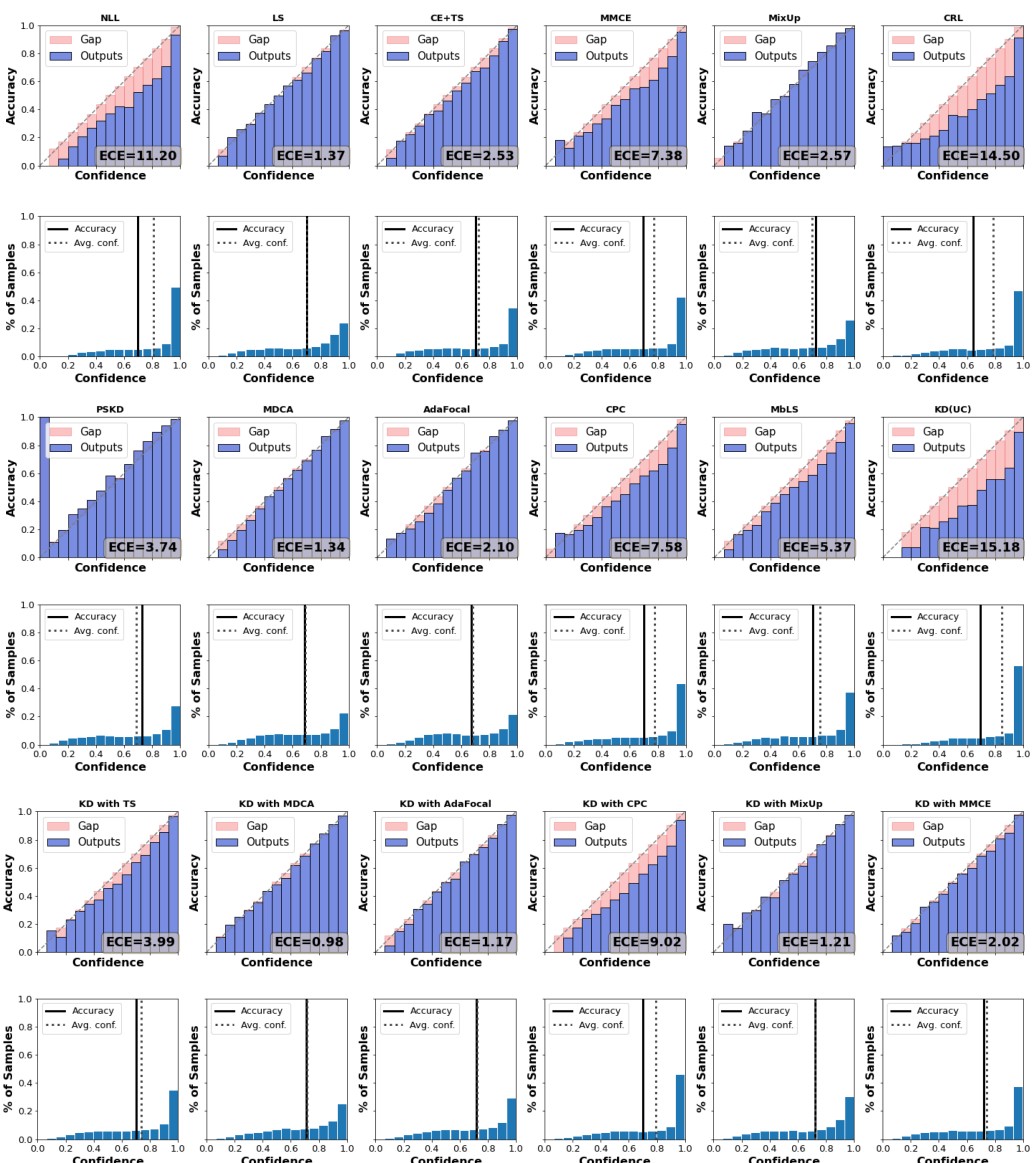

Figure S6: Reliability Plots for `top-5` KD with **(Ours)** techniques on WideResNet-40-1 on `CIFAR100`. Teacher used: WideResNet-40-2. **KD(C)** framework achieves competitive calibration results for `KD with MDCA`, `KD with AdaFocal` and `KD with MixUp`.

(2021), which suggest that maximizing fidelity with the teacher model yields the best transfer of properties.

## 5.3 CALIBRATION PERFORMANCE UNDER DATASET DRIFT

`DNNs` are found to be over-confident and highly uncalibrated under dataset/domain shift Tomani et al. (2020). We investigate the robustness of our method KD(C) by examining the degradation in calibration under natural/non-semantic shift (images with the same label but different distribution). We carry out this study for `ResNet56` pre-trained on the `CIFAR100` dataset along with various calibration techniques and report the evaluation results on `CIFAR100-C` Hendrycks & Gimpel (2016) in Fig. S9. We used `ResNet56` models that were trained with `ResNet110` as teacher

| Calibration Method | Top1 (%) ↑ | ECE (%) ↓ | SCE (%) ↓ | AECE (%) ↓ |
|---|---|---|---|---|
| NLL | 50.43 | 13.72 | 0.24 | 13.72 |
| LS Szegedy et al. (2015) | 51.20 | 3.84 | **0.19** | 3.88 |
| CE with TS Guo et al. (2017) | 50.43 | 13.72 | 0.24 | 13.72 |
| MMCE Kumar et al. (2018) | 50.30 | 11.32 | 0.21 | 11.32 |
| MixUp Thulasidasan et al. (2019) | 52.02 | 4.74 | **0.19** | 4.73 |
| PSKD Kim et al. (2021) | **53.66** | 13.27 | 0.21 | 13.27 |
| MDCA Hebbalaguppe et al. (2022) | 46.81 | 1.52 | **0.19** | **1.11** |
| CPC Cheng & Vasconcelos (2022) | 51.27 | 12.01 | 0.21 | 12.01 |
| MbLS Liu et al. (2022) | 50.11 | 8.87 | 0.20 | 8.87 |
| KD(UC) | 49.31 | 4.20 | 0.20 | 4.20 |
| **Ours (KD with MDCA)** | 45.79 | **0.85** | 0.21 | 1.17 |
| **Ours (KD with LS)** | 49.69 | 2.69 | **0.19** | 2.68 |
| **Ours (KD with MbLS)** | 49.33 | 2.86 | 0.20 | 2.89 |

Table T1: **[Self-distillation]** using MobileNetV2 feature extractor on `Tiny-ImageNet` dataset. Note that the main paper reported self-distillation results using the MobileNetV2 feature extractor on the `CIFAR10` dataset. `Top3` best KD(C) variants are reported. `KD with MDCA` variant of `KD(C)` achieve competitive calibration results with `SOTA`.

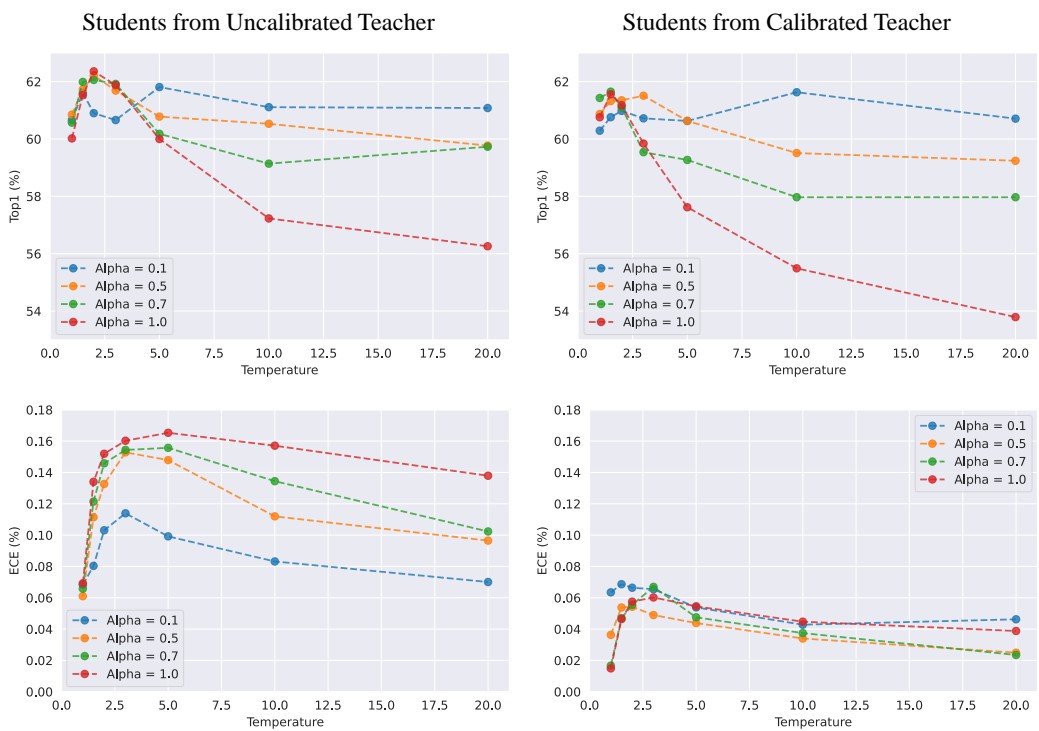

Figure S7: We study the effect of varying temperature, $T$ and distillation weight $\alpha$, on `ECE` and top $1\%$ accurracy when `ResNet56` teacher model is used and `ResNet8` as student on `CIFAR100` dataset. Observe the optimal values of `ECE` and top $1\%$ accuracy when $T$ is set around 1. For calibration `KD with MDCA` was used.

for KD(UC) and KD(C). We observe KD(UC) and KD(C) achieve the highest accuracy across all severities, with the latter achieving close to the best `ECE` (`LS` achieves best `ECE`), however, KD(C) achieves the best `AUROC` score in comparison to any other calibration technique. This indicates, KD(C) is better across all metrics measuring reliability (be it calibration or refinement, while also giving an additional boost in accuracy).

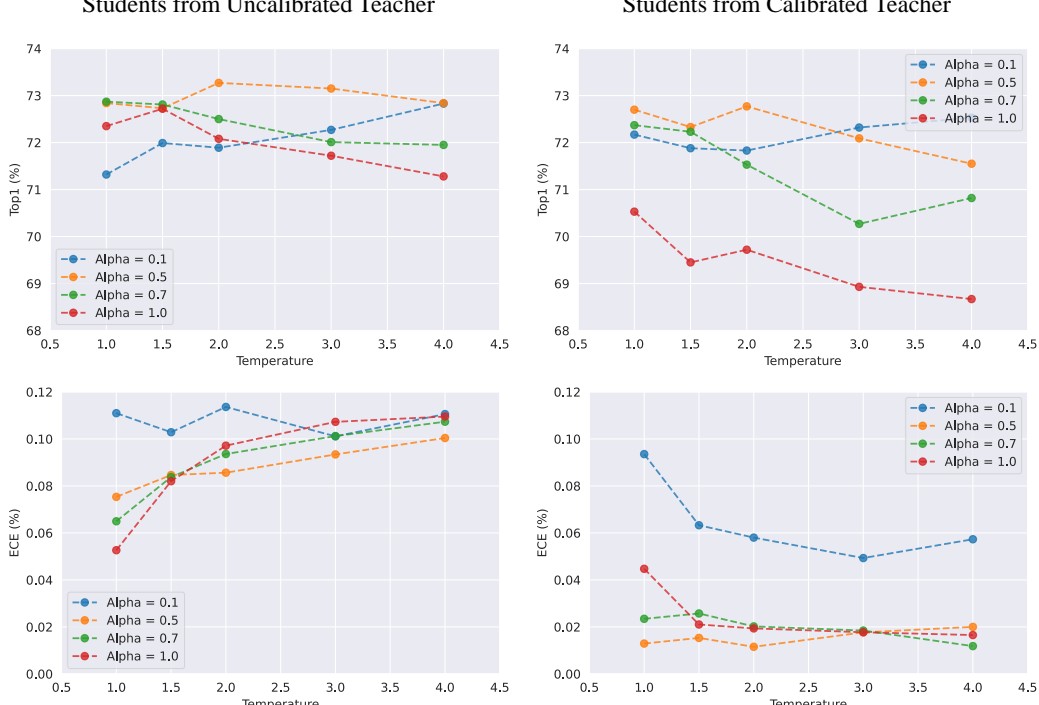

Figure S8: We study the effect of varying temperature, $T$ and distillation weight $\alpha$, on `ECE` and top $1\%$ accuracy when `ResNet32` teacher model is used and `ResNet56` as student on `CIFAR100` dataset. `KD with MDCA` was used for calibration.

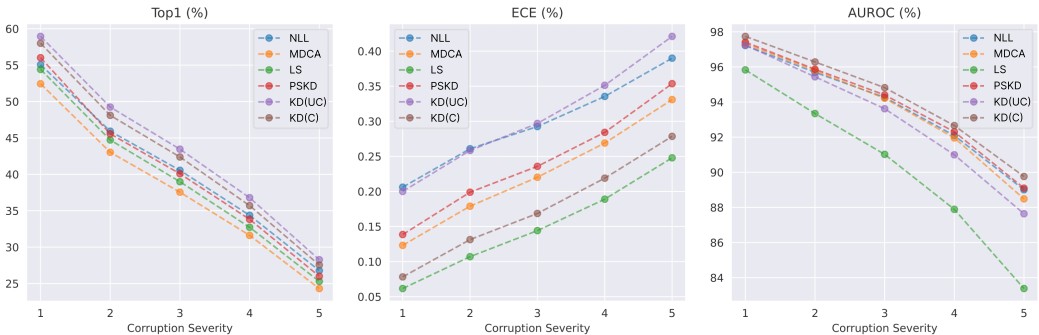

Figure S9: Robustness to corruption, tested on `CIFAR100-C` dataset Hendrycks & Gimpel (2016) using `ResNet-56`. KD(UC) and KD(C) were trained using `ResNet-110` as a Teacher. Note that KD(C) provides a good trade-off between accuracy and calibration, at the same time achieving the highest AUROC (even though LS outperforms KD(C) by a tiny margin in terms of calibration, KD(C) has significantly better AUROC and accuracy. AUROC indicates better inter-class separability in classifiers thereby enhancing trustworthiness in addition to calibration benefits. KD(C) uses `KD with MDCA` variant.

## 6 TRAINING DETAILS

In this section, we provide a detailed summary of the hyperparameters and training techniques used, in order to ensure reproducibility. All models have been trained on 40GB Nvidia A100 GPUs.

| Dataset | Teacher Calibration | Teacher | Top1 (%) | ECE (%) | SCE (%) | ACE (%) | Student | Top1 (%) | ECE (%) | SCE (%) | ACE (%) |
|---|---|---|---|---|---|---|---|---|---|---|---|
| CIFAR100 | NLL | WRN-40-2 | 74.10 | 13.42 | 0.32 | 13.42 | WRN-40-1 | 69.60 | 15.18 | 0.37 | 15.18 |
| | LS | | 74.67 | 2.44 | 0.21 | 2.21 | | 70.54 | 1.22 | 0.21 | 1.20 |
| | CE+TS | | 74.10 | 2.18 | 0.20 | 2.15 | | 70.07 | 4.00 | 0.21 | 4.00 |
| | MMCE | | 73.04 | 5.21 | 0.21 | 5.14 | | 72.08 | 2.02 | 0.19 | 1.95 |
| | MixUp | | 77.46 | 1.50 | 0.19 | 1.43 | | 72.48 | 1.21 | 0.20 | 1.17 |
| | CRL | | 70.71 | 12.20 | 0.32 | 12.11 | | 69.76 | 7.21 | 0.23 | 7.04 |
| | MDCA | | 73.74 | 1.36 | 0.19 | 1.26 | | 71.07 | 0.98 | 0.20 | 1.10 |
| | AdaFocal | | 73.24 | 2.43 | 0.20 | 2.31 | | 71.70 | 1.19 | 0.19 | 1.34 |
| | CPC | | 75.09 | 11.06 | 0.28 | 10.99 | | 70.00 | 9.02 | 0.26 | 9.01 |
| | MbLS | | 73.54 | 5.53 | 0.22 | 5.50 | | 71.44 | 3.70 | 0.22 | 3.41 |
| | NLL | RNXT-18x4 | 63.69 | 17.20 | 0.41 | 17.20 | MNV2 | 66.82 | 5.40 | 0.22 | 5.36 |
| | LS | | 63.89 | 5.00 | 0.25 | 5.37 | | 66.63 | 2.48 | 0.24 | 2.50 |
| | CE+TS | | 63.69 | 2.73 | 0.23 | 2.74 | | 67.22 | 1.63 | 0.19 | 1.58 |
| | MMCE | | 62.40 | 6.53 | 0.24 | 6.56 | | 66.24 | 1.47 | 0.19 | 1.64 |
| | MixUp | | 65.57 | 3.15 | 0.24 | 3.17 | | 69.92 | 2.17 | 0.24 | 2.10 |
| | CRL | | 52.98 | 20.21 | 0.51 | 20.21 | | 64.17 | 2.89 | 0.23 | 2.79 |
| | MDCA | | 63.70 | 2.14 | 0.22 | 2.23 | | 67.17 | 1.10 | 0.20 | 1.17 |
| | AdaFocal | | 64.55 | 6.19 | 0.24 | 6.15 | | 66.64 | 1.55 | 0.20 | 1.43 |
| | CPC | | 63.20 | 8.98 | 0.28 | 8.97 | | 67.83 | 0.88 | 0.19 | 0.95 |
| | MbLS | | 64.42 | 12.89 | 0.32 | 12.88 | | 67.50 | 3.21 | 0.20 | 3.22 |
| CIFAR10 | NLL | MNV2 | 89.87 | 3.30 | 0.75 | 3.28 | DN | 90.2 | 2.17 | 0.60 | 2.13 |
| | LS | | 89.60 | 7.10 | 1.78 | 6.75 | | 92.65 | 4.27 | 1.23 | 4.13 |
| | CE+TS | | 89.90 | 0.98 | 0.40 | 0.77 | | 93.25 | 0.62 | 0.43 | 0.54 |
| | MMCE | | 89.38 | 1.20 | 0.51 | 0.94 | | 92.13 | 0.81 | 0.45 | 0.69 |
| | MixUp | | 89.57 | 9.42 | 2.07 | 9.41 | | 91.19 | 5.20 | 1.33 | 5.01 |
| | CRL | | 90.31 | 2.92 | 0.72 | 2.81 | | 92.24 | 2.58 | 0.68 | 2.54 |
| | MDCA | | 88.74 | 0.99 | 0.46 | 0.80 | | 90.90 | 0.53 | 0.45 | 0.51 |
| | AdaFocal | | 88.98 | 0.79 | 0.44 | 0.86 | | 91.68 | 0.54 | 0.34 | 0.61 |
| | CPC | | 89.26 | 3.47 | 0.79 | 3.44 | | 93.14 | 0.74 | 0.43 | 0.85 |
| | MbLS | | 89.86 | 2.83 | 0.69 | 2.78 | | 93.1 | 0.61 | 0.38 | 0.40 |

Table T3: Comparison of evaluation metrics of Teacher-Student pairs. Observe that calibration transfer takes place from a calibrated teacher more or less to a student. The minor differences of calibration values can be attributed to the capacity gap between the teacher and student. WRN: WideResNet, RNXT: ResNeXt, DN: DenseNet. Number of parameters: WRN-40-1: 0.56M ; WRN-40-2: 2.24M ; MNV2: 2.25M ; RNXT-18x4: 25.46M ; DN: 6.95M

The code was written using the PyTorch framework. We make use of automatic mixed precision training in order to reduce training time. We borrow some code from the official implementation of Hebbalaguppe et al. (2022); Mukhoti et al. (2020); Yuan et al. (2021).

For `CIFAR10/100` datasets, we train all `ResNets / WideResNets / ResNeXt-18x4` models using a learning rate of $0.1$ for 160 epochs. The learning rate is decayed by a factor of 10 at epoch 80 and 120. We use SGD optimizer with momentum 0.9 and weight decay of $5e-4$. We use a batch size of 128. For the larger models like `ResNet-110`, we train them using a learning rate of $0.05$ for 240 epochs. The learning rate is decayed by a factor of 10 at epoch 150, 180 and 210. We use SGD optimizer with momentum 0.9 and weight decay of $5e-4$. We use a batch size of 64. ConvNet2 model has been trained just like all other models, except for the learning rate which is set to 0.01, without a learning rate decay scheduler.

For `Tiny-ImageNet` dataset, all models are trained using a maximum learning rate of $0.1$ with a cosine annealing learning rate with a warmup of 1000 steps with minimum learning rate $1e-5$. The weight decay and momentum are $5e-4$ and $0.9$ respectively. We train the models for 100 epochs with a batch size of 128.

For training students using KD, we use the same hyper-parameters for the respective datasets. For big-to-small KD (e.g. `WideResNet-40-2` → `WideResNet-40-1`), we grid search $T$ (temperature) and $\alpha$ (distillation weight) in the ranges $\{1, 1.5, 2, 3, 4, 5, 10, 20\}$ and $\{0.9, 1.0\}$ respectively. For small-to-big KD and self-distillation (e.g. `MobileNetV2` ↓ `DenseNet-121`, `MobileNetV2` → `MobileNetV2`), we grid search $T$ and $\alpha$ in the ranges $\{1, 1.5, 2, 3, 4\}$ and $\{0.1, 0.3, 0.5, 0.7, 0.9, 1.0\}$ respectively.

For baselines, we use the recommended hyperparameters as suggested by the respective authors Hebbalaguppe et al. (2022); Szegedy et al. (2015); Kim et al. (2021); Kumar et al. (2019); Cheng & Vasconcelos (2022); Thulasidasan et al. (2019); Liu et al. (2022); Moon et al. (2020); Ghosh et al.

| Dataset | Teacher Model | Student Model | Temperature | Top1 (%) | ECE (%) | SCE (%) | ACE (%) |
|---|---|---|---|---|---|---|---|
| CIFAR100 | WRN-40-2 | WRN-40-1 | 0.10 | 71.06 | 26.40 | 0.55 | 26.39 |
| | | | 0.20 | 71.06 | 23.79 | 0.52 | 23.79 |
| | | | 0.50 | 71.06 | 15.74 | 0.38 | 15.74 |
| | | | 0.75 | 71.07 | 8.44 | 0.27 | 8.44 |
| | | | **1.00** | **71.06** | **0.98** | **0.20** | **1.10** |
| | | | 1.25 | 71.06 | 8.60 | 0.27 | 8.60 |
| | | | 1.50 | 71.06 | 18.00 | 0.42 | 18.00 |
| | | | 1.75 | 71.06 | 27.11 | 0.58 | 27.11 |
| | | | 2.00 | 71.06 | 35.22 | 0.74 | 35.22 |
| | | | 2.25 | 71.06 | 41.99 | 0.88 | 41.99 |
| | | | 2.50 | 71.06 | 47.39 | 0.99 | 47.39 |
| | | | 2.75 | 71.06 | 51.60 | 1.07 | 51.60 |
| | | | 3.00 | 71.06 | 54.86 | 1.12 | 54.86 |
| | | | 3.25 | 71.06 | 57.37 | 1.13 | 57.37 |
| | | | 3.50 | 71.06 | 59.32 | 1.11 | 59.32 |
| | | | 3.75 | 71.06 | 60.86 | 1.07 | 60.86 |
| | | | 4.00 | 71.06 | 62.08 | 1.00 | 62.08 |
| | | | 4.25 | 71.06 | 63.06 | 0.92 | 63.06 |
| | | | 4.50 | 71.06 | 63.86 | 0.81 | 63.86 |
| | | | 4.75 | 71.06 | 64.52 | 0.69 | 64.52 |
| | | | 5.00 | 71.06 | 65.07 | 0.56 | 65.07 |
| | RNXT-18x4 | MNV2 | 0.10 | 67.17 | 29.85 | 0.63 | 29.85 |
| | | | 0.20 | 67.17 | 26.81 | 0.58 | 26.80 |
| | | | 0.50 | 67.17 | 17.33 | 0.41 | 17.33 |
| | | | 0.75 | 67.17 | 8.96 | 0.28 | 8.96 |
| | | | **1.00** | **67.17** | **1.10** | **0.20** | **1.17** |
| | | | 1.25 | 67.18 | 9.60 | 0.28 | 9.60 |
| | | | 1.50 | 67.18 | 19.24 | 0.44 | 19.24 |
| | | | 1.75 | 67.17 | 28.23 | 0.62 | 28.23 |
| | | | 2.00 | 67.17 | 35.97 | 0.78 | 35.97 |
| | | | 2.25 | 67.17 | 42.21 | 0.91 | 42.21 |
| | | | 2.50 | 67.17 | 47.04 | 1.00 | 47.04 |
| | | | 2.75 | 67.17 | 50.70 | 1.05 | 50.70 |
| | | | 3.00 | 67.18 | 53.48 | 1.06 | 53.48 |
| | | | 3.25 | 67.18 | 55.59 | 1.05 | 55.59 |
| | | | 3.50 | 67.17 | 57.20 | 1.01 | 57.20 |
| | | | 3.75 | 67.17 | 58.47 | 0.94 | 58.47 |
| | | | 4.00 | 67.17 | 59.47 | 0.85 | 59.47 |
| | | | 4.25 | 67.17 | 60.27 | 0.75 | 60.27 |
| | | | 4.50 | 67.17 | 60.92 | 0.64 | 60.92 |
| | | | 4.75 | 67.17 | 61.46 | 0.52 | 61.47 |
| | | | 5.00 | 67.17 | 61.92 | 0.40 | 61.92 |

Table T4: **[Effect of `TS` on KD(C)]:** The student is calibrated by distilling from an `MDCA` calibrated teacher `KD with MDCA` (a variant of KD(C)). The table shows that further temperature scaling (`TS`) does not impact the models trained with `KD with MDCA` as they are calibrated to start with. Parameters: WRN-40-1: 0.56M ; WRN-40-2: 2.24M ; MNV2: 2.25M ; RNXT-18x4: 25.46M

(2022), i.e. for LS Szegedy et al. (2015), we use smoothing of 0.1; for PSKD Kim et al. (2021) we use $\alpha = 0.8$; for MixUp Zhang et al. (2018) the mixup hyperaparameter was taken as 0.4 as it was reported to be the best by the authors, for MDCA Hebbalaguppe et al. (2022), we grid search for the best performing $\beta \in \{1, 5, 10\}$ and $\gamma \in \{1, 2, 3, 4, 5\}$; For MMCE, we grid search for the best performing $\beta \in \{1, 2, 3, 4, 5\}$.

We provide the metrics for various teachers trained from scratch and used throughout the paper in Tab. T3. These teachers have been used to train various student models for KD(UC) and KD(C) variants mentioned in the paper.

# 7  REPRODUCIBILITY

In the spirit of reproducible research, we intend to make the source code available post-acceptance. To aid reviewers, the source code for our approach is attached along with the supplemental material. Details of our setup and implementation of the baselines can be found at: `Code/README.md` folder.

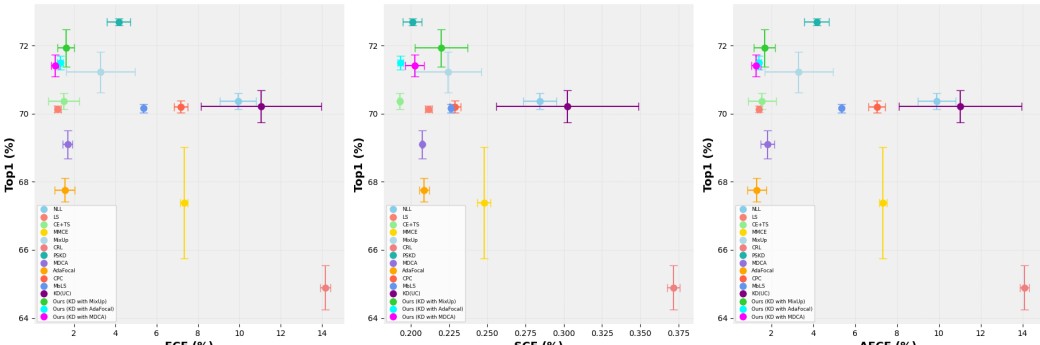

Figure S7: **Comparative study of accuracy vs. calibration trade-offs associated with existing calibration techniques and ours (Top-left is most preferred)**: The mean and one standard scatter error bars for `Top1`, `ECE` and `SCE` of `WideResNet-40-1` trained on `CIFAR100` using `SOTA` calibration techniques. `WideResNet-40-2` was used as Teacher for KD(UC) and the proposed, KD(C) variants. Note: KD(C) variants (magenta, cyan, and green) achieve the best results in terms of `ECE`, `ACE` and `SCE`, along with slight boosts in `Top1` (an inherent KD-property). Further, the lower variances emphasize the reliability of KD(C) variants. All plots were generated by training `WideResNet-40-1` models through every calibration technique on 3 runs.

# 8 LIMITATIONS

While our work paves way to create optimal lightweight models that are both accurate and calibrated, it is important to acknowledge three potential limitations that we plan to address in future research - (a) principled approach to select hyperparameters, such as the temperature $T$, distillation weight $\alpha$, calibration regularization coefficient $\beta$, and characterization of optimal student-teacher capacity difference for best calibration, (b) extending theoretical insights to general nonlinear networks, (c) benchmarking KD(C) on natural language processing (NLP) tasks, particularly when the teacher networks belong to the family of large language models (LLMs). This is particularly challenging due to the unavailability of adequate computational resources.

# 9 BROADER IMPACT

Bigger DNN models aren't necessarily better models. From a deployment standpoint, the size of the weights affects the inference time and storage constraints on edge devices which is crucial in applications such as augmented reality and robotics. Our proposed algorithm has the potential to be employed in trustworthy lightweight models on the edge. In our endeavor to deploy lightweight models that are also reliable, we delve into the realm of knowledge distillation, extending its traditional function of transferring accuracy from teacher networks to student networks. Through this exploration, we have discovered a novel approach to calibrating models effectively. We present, arguably for the first time, compelling evidence that model calibration can be achieved without sacrificing accuracy through knowledge distillation. Notably, our implementation of knowledge distillation not only guarantees enhanced model calibration but also outperforms the accuracy obtained through conventional training from scratch in specific cases. This innovative approach enables us to simultaneously accomplish the dual objectives of optimal calibration and improved accuracy.

Towards this end, we provide extensive theoretical findings that extend beyond the realms of accuracy transfer and calibration alone. We show, through optics of linear teacher and student networks, that the optimization of student network weights through knowledge distillation enables them to exhibit similar behavior and performance as their respective teachers (see Theorem 1 in the main text). Subsequently, the scope of producing trustworthy models can also be extended to incorporate characteristics, such as fairness and refinement. On a more specific note, Theorem 2 in our work shows that there is a definite advantage of working with calibrated teachers over uncalibrated teachers, i.e., calibrated teachers tend to produce calibrated students without compromising on accuracy. Hence

our approach, KD(C), centers around train-time calibration of teacher models, enabling them to generate accurate and optimally calibrated students through knowledge distillation. Significantly, based on our empirical evaluations, it is evident that the transfer of calibration operates bidirectionally. This means that larger calibrated models can be utilized to create smaller calibrated models, and conversely, smaller calibrated models can also serve as a foundation for generating larger calibrated models.

Overall, the research contributes to the advancement of model calibration, accuracy, trustworthiness, and scalability, which can have significant implications in various fields relying on the deployment of reliable and lightweight models.

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
