# OpenReview forum: "Understanding Calibration Transfer in Knowledge Distillation"
_ICLR.cc/2024/Conference — ICLR 2024 Conference Withdrawn Submission_

### Official Review · Reviewer_U8KR · 2023-10-19

**Soundness:** 3 good
**Presentation:** 3 good
**Contribution:** 2 fair
**Rating:** 5
**Confidence:** 4

**Summary:**

This paper provide a robust theoretical foundation, including formal proofs, for the transfer of calibration and accuracy between teacher and student DNN models. The authors shows consistent superior calibration achieved through our KD(C) method.

**Strengths:**

KD and calibration has a strong relationship, where both work on the output logits. It is good to see a work connect them together.

**Weaknesses:**

See questions.

**Questions:**

1. For results in Table 2: From my understanding, the ECE value has a variance around 1%, how many times do you run to get the results reported in the table 2?
2. The accuracy can affect the ECE a lot. The accuracy of WideResNet-40-1 (0.56M) and MobileNetV2 (2.25M) on CIFAR100 seems not as good as the PyTorch default version.
3. I would like to see the discussion about distill from transformer to CNN since it is said transformers are normally better calibrated [Revisiting the Calibration of Modern Neural Networks]
4. [Dual Focal Loss for Calibration] seems achieve the same results as yours on CIFAR10. Will KD improved upon dual focal loss?

---

### Official Review · Reviewer_p7Qx · 2023-10-28

**Soundness:** 2 fair
**Presentation:** 3 good
**Contribution:** 1 poor
**Rating:** 3
**Confidence:** 4

**Summary:**

This paper is about knowledge distillation and its relation to calibration. The authors argue that to produce calibrated student models, a calibrated teacher model is required.

Contributions are:
- A theoretical framework using linear networks to explore and understand knowldge distillation from a calibration perspective.
- Experiments showing that with deep neural networks, students are best calibrated when the teacher is calibrated too, and this is the best calibration when compared to train-time and post-hoc calibration methods. The proposed method is called Knowledge Distillation from Calibrated teacher (KDC).
- Results showing that calibration distillation works both from large to small models and small to large models, and in iterative self-distillation.

**Strengths:**

- The paper is well written and clear to understand, minus the proofs I mention below.
- The topic is of broad interest, knowledge distillation is used in practice to train smaller models and transferring some knowledge from large models, for example at the edge where large compute is not available, and often calibration is ignored, specially when using label smoothing. This paper contributes with insights and results that show that calibration does also transfer from teacher to student, when the teacher is better calibration, and in other similar settings too.
- The experimental setup seems correct, experiments are done on CIFAR10 and CIFAR100 with state of the art convolutional networks with a good selection of calibration metrics. The selection of baselines is okay but I note in weaknesses that there are at least three papers that intersect with this work and are possible baselines.
- Experimental results show that KDC with different calibration methods applied to the teacher network, does distill a calibrated student from the calibrated teacher network. This applies also to small teachers to large students, and iterative self-distillation. So there is evidence that the theoretical framework for linear networks also empirically extends into deep networks.

**Weaknesses:**

- I have reviewed the proof and formulation for Theorem 2, which aims to prove that calibrated teachers lead to calibrated students, but I do not understand the criteria in Equation 3. Calibration is measured through a reliability plot, and by comparing empirical confidences with empirical accuracies, I do not see this in the formulation in Equation 3, and its not clear why the previous definition for calibration would extend to Equation 3. From what I can interpret in this equation, it is summing the probabilities of the teacher and comparing them to the sum of the labels, which does not directly correspond to a zero calibration error. I would need clarification or further explanation here, specially as the whole proof is made starting from the formulation in Equation 3.

- I am not sure of the novelty of this paper, there are several papers that explore knowledge distillation and calibrated teacher models:

Guo H, Pasunuru R, Bansal M. An overview of uncertainty calibration for text classification and the role of distillation. InProceedings of the 6th Workshop on Representation Learning for NLP (RepL4NLP-2021) 2021 Aug (pp. 289-306).

Gurau C, Bewley A, Posner I. Dropout distillation for efficiently estimating model confidence. arXiv preprint arXiv:1809.10562. 2018 Sep 27.

Bulò SR, Porzi L, Kontschieder P. Dropout distillation. InInternational Conference on Machine Learning 2016 Jun 11 (pp. 99-107). PMLR.

These papers all train teacher models with some uncertainty estimation method (MC-Dropout, Ensembles, etc), and then distill  the teacher model (which due to explicit uncertainty estimation methods, is better calibrated) into a student model without uncertainty that is also often better calibrated. While these papers do not use the same calibration methods for the teacher model, they do proof that calibration is transferred through knowledge distillation, which I believe is one of the major claims of this paper.

I also note that these papers are not cited in the current version of this paper. I believe two things: these papers could be baselines, or from these papers it can be inferred that a calibrated teacher leads to a calibrated student model. This paper makes this exact claim, while the previous three papers I mention here, do not make that claim directly, but it is implied from their results. They basically say that distilling a teacher model with uncertainty estimation, will also transfer that uncertainty estimation to the student model, which is basically the same claim made in this paper.

I believe that these three papers (there could be more) severely weaken the novelty of this paper.

Minor Comments
- If you could not experiment on Tiny ImageNet, then there is no point on mentioning this dataset in the experimental setup.

**Questions:**

- Can you further explain the formulation in Equation 3? How can we infer that probabilities are calibrated by summing those probabilities and comparing them to the sum of the ground truth labels?
- Considering the three papers I have mentioned in weaknesses, what is the novelty of your paper compared to those papers?

---

### Official Review · Reviewer_VrHg · 2023-10-28

**Soundness:** 3 good
**Presentation:** 3 good
**Contribution:** 3 good
**Rating:** 6
**Confidence:** 4

**Summary:**

The problem of calibrating deep neural networks is considered. This work investigates the effect of knowledge distillation (KD) on calibration. Some interesting observations are provided: Calibration can be transferred from small-scale teacher network to large-scale student network; Compatibility with existing proposed calibration training techniques cannot always be assumed.
Finally, the proposed KD-based calibration training approach is shown to achieve state-of-the-art calibration on CIFAR-100.

**Strengths:**

1.	This work provides the observation that knowledge distillation is another potential direction for calibrating DNNs in addition to the existing approaches. Also, some theoretical insight on why knowledge distillation can help calibration are presented.
2.	By the combination of KD and other train/test-time calibration approach, state-of-the-art calibration performance is achieved on CIFAR100 with WRN-40-1 and MobileNetV2.

**Weaknesses:**

1.	The empirical evaluation seems to be limited on the CIFAR100, which is small-scale and arguably less challenging than most real-world datasets. The contribution of the work should be better validated on more realistic data (e.g. ImageNet) or more challenging setting (for example under data corruption and perturbation, e.g. CIFAR-100-C [1]). Also, all the evaluated network architectures are convolutional neural networks, however, the transformer-based DNNs are becoming more and more popular in vision and language communities. The missing evaluation of a transformer-based network in KD for calibration is of more interests and relevance to today’s community.

[1] Dan Hendrycks and Thomas Dietterich. Benchmarking neural network robustness to common corruptions and perturbations. In ICLR, 2018.

**Questions:**

1.	In table 2, an odd observation is that KD w/ CPC achieves best calibration with MobileNetV2. But this not the case of WRN-40-1. On the contrary, the calibration error with WRN-40-1 is significantly worse most other evaluated baselines. I am curious if there is any possible explanation for the inconsistent observations.

---

### Official Review · Reviewer_J1h8 · 2023-11-02

**Soundness:** 2 fair
**Presentation:** 2 fair
**Contribution:** 2 fair
**Rating:** 3
**Confidence:** 4

**Summary:**

This paper studies the uncertainty calibration of neural networks under knowledge distillation (KD) framework. First, the authors give a theoretic understanding calibration with KD under the assumption of linear classifier, which says that a well-calibrated teacher can distill a well-calibrated student. Then, the authors give an empirical study of calibration with KD, which involves several training-time calibration techniques. The experiments show that with a well-calibrated teacher, which can be either larger or smaller than the student model, KD can produce student with good calibration than many training-time/post-hoc calibration methods.

**Strengths:**

1. The studied problem is interesting and important. One interesting point of KD for calibration is that KD attempts to change the hard labels in datasests to soft labels, which may correct the noisy supervision for some ambiguous examples and improve calibration for student models.

2. From theoretic perspective, this paper proves that the student models distilled from a well-calibrated teacher are also well-calibrated under the linear classifier assumption. From the empirical perspective, the experiments show some positive results of the help of KD for calibration.

**Weaknesses:**

1. The theoretic results are based on a special class of linear teacher and student classifiers. As the KD framework are usually studied in deep learning paradigm, the theoretic analysis based linear classifier is kind of weak for understanding KD in deep learning. Although the authors argue that the input of the linear classifiers can be the features extracted from deep backbones, this results still cannot fully explain why KD helps calibration since the representation learning in the KD framework is not considered in this paper. Furthermore, the miscalibration (over-confidence) issue is widely considered as a more challenge problem in deep models rather than linear models.

2. The experimental setup is somewhat confusing. For the comparison methods like Mixup, AdaFocal, CPC and MDCA, their hyperparameters were used as recommended in their original papers. However, for the proposed method, it is unclear whether (setting 1) the hyperparameters for the teacher model were set identically with the baselines, or (setting 2) if they were tuned on the validation set to find the optimal parameters that minimized the calibration error. If setting 1 is adopted, it is not consistently with the motivation stated in this paper that the teacher model is well-calibrated, since it is clear that the calibration performance is not good as shown in the top rows in Table 2. If setting 2 is adopted, the comparison between the proposed method and the baselines (e.g., KD with Mixup vs Mixup) is not fair, since the hyperparameters of baselines are not optimized in terms of the calibration error.

3. The statement of "post-hoc methods are known to result in relatively inferior calibration performance" is not well supported by the literature in calibration. For example, previous works [1][2][3] show the effectiveness of post-hoc calibration methods even compared with training-time calibration methods or the integration of training-time calibration methods and post-hoc calibration techniques. The results in Table 2 in this paper also show that simple CE+TS is better than many training-time calibration methods.

[1] On calibration of modern deep neural networks, ICML 2017
[2] On the pitfalls of mixup on uncertainty calibration, CVPR 2023
[3] A benchmark study of calibration, Arxiv 2023

**Questions:**

Please refer to the weaknesses section.